# ERRγ enhances cardiac maturation with T-tubule formation in human iPSC-derived cardiomyocytes

Kenji Miki [1,2,7], Kohei Deguchi[2,3,7], Misato Nakanishi-Koakutsu[1,2,7], Antonio Lucena-Cacace [1], Shigeru Kondo[2,3], Yuya Fujiwara[1,2], Takeshi Hatani [1], Masako Sasaki[1,2], Yuki Naka[1,2], Chikako Okubo [1], Megumi Narita [1], Ikue Takei[1,2], Stephanie C. Napier [2,3], Tsukasa Sugo[4], Sachiko Imaichi[5], Taku Monjo[6], Tatsuya Ando[6], Norihisa Tamura[2,3], Kenichi Imahashi[2,3], Tomoyuki Nishimoto[2,3 ✉] & Yoshinori Yoshida [1,2 ✉]

One of the earliest maturation steps in cardiomyocytes (CMs) is the sarcomere protein isoform switch between TNNI1 and TNNI3 (fetal and neonatal/adult troponin I). Here, we generate human induced pluripotent stem cells (hiPSCs) carrying a TNNI1^EmGFP and TNNI3^mCherry double reporter to monitor and isolate mature sub-populations during cardiac differentiation. Extensive drug screening identifies two compounds, an estrogen-related receptor gamma (ERRγ) agonist and an S-phase kinase-associated protein 2 inhibitor, that enhances cardiac maturation and a significant change to TNNI3 expression. Expression, morphological, functional, and molecular analyses indicate that hiPSC-CMs treated with the ERRγ agonist show a larger cell size, longer sarcomere length, the presence of transverse tubules, and enhanced metabolic function and contractile and electrical properties. Here, we show that ERRγ-treated hiPSC-CMs have a mature cellular property consistent with neonatal CMs and are useful for disease modeling and regenerative medicine.

[1] Center for iPS Cell Research and Application, Kyoto University, Kyoto, Japan. [2] Takeda-CiRA Joint program (T-CiRA), Fujisawa, Japan. [3] Regenerative Medicine Unit, Takeda Pharmaceutical Company Limited, Fujisawa, Japan. [4] GenAhead Bio Inc., Fujisawa, Japan. [5] Pharmaceutical Science, Takeda Pharmaceutical Company Limited, Fujisawa, Japan. [6] IBL, Takeda Pharmaceutical Company Limited, Fujisawa, Japan. [7] These authors contributed equally: Kenji Miki, Kohei Deguchi, Misato Nakanishi-Koakutsu. ✉email: tomoyuki.nishimoto@takeda.com; yoshinor@cira.kyoto-u.ac.jp

During heart development, cardiac transcription factors regulate the terminal differentiation to cardiac-specific subtypes. In principle, human-induced pluripotent stem cells (hiPSCs)[1] can differentiate into these subtypes and their intermediates[2,3]. However, the maturation and functionality of the differentiated cells are neither optimal nor understood[4–6], resulting in a heterogeneous pool of atrial, nodal, and ventricular fetal-like cardiomyocytes (CMs).

Several molecular events define the maturation of the myocardium[7–9]. Irreversible isoform switches in sarcomere protein troponin I (TNNI) represent a reliable method to monitor cardiac maturation during development[10–13]. Specifically, isoform switching between TNNI1 (slow skeletal TnI; ssTnI) and TNNI3 (cardiac TnI; cTnI) occurs during the developmental transition from fetal to neonatal and postnatal stages and is followed by the eventual acquisition of functional hallmarks of cardiac maturation[7]. These hallmarks include an efficient conversion of energy, excitation–contraction coupling, a positive force–frequency relationship, anisotropy driven by the cellular alignment, and neonatal/adult CM-like electrophysiological properties with increased conduction velocity.

Along with structural maturation via the TNNI isoform switch[7], metabolic maturation occurs; the neonatal and adult myocardia can meet the new metabolic requirements after birth[14]. In the fetal heart, 50% of energy production in the form of ATP derives from glycolysis, and only 15% of the total energy production is derived from fatty acids. This metabolism is driven by the low oxygen environment of the mammalian utero, which has high levels of lactate and low levels of circulating fatty acids to favor high glycolytic activity.

Finally, the expression of cell cycle regulators, such as cyclins, cyclin-dependent kinases and cyclin-dependent kinase inhibitors, changes dynamically during cardiogenesis and heart development[15,16]. As a consequence, postnatal CMs in vivo display cell-cycle arrest, leading to a loss of proliferative capacity and regenerative potential.

In this work, we identify mature subpopulations from hiPSCs differentiated to CM lineages (hiPSC-CMs) using a TNNI1[EmGFP] and TNNI3[mCherry] hiPSC double reporter to detect early-stage phenotypes of cardiac maturation. Extensive drug screening identifies two compounds that can enhance cardiac maturation through a scalable culture method leading to significant changes in TNNI3 expression: an estrogen-related receptor gamma (ERRγ) agonist and an S-phase kinase-associated protein 2 (SKP2) inhibitor. Subsequent functional validation shows that the ERRγ agonist also enhances the metabolic maturation to reduce glycolysis dependency. Altogether, our results indicate that ERRγ activation plays an important function in the fundamental biology of cardiac maturation.

## Results

### Generation of the hiPSC TNNI1 and TNNI3 dual reporter.
To delineate mature CM lineages, we developed a hiPSC double reporter expressing emerald green fluorescent protein (EmGFP) and mCherry from the loci of TNNI1 and TNNI3, respectively, using CRISPR-Cas9 (Fig. 1a and b). The EmGFP and mCherry elements were placed downstream of TNNI1 and TNNI3, respectively. The protein level of TNNI1 and TNNI3 in hiPSC-CMs was comparable between the reporter and parental line. A single hiPSC clone was identified as a recombinant cell line in which 2A-EmGFP was inserted into the region of the TNNI1 stop codon (Supplementary Fig. 1a and b). Once the selection cassette was removed (Fig. 1c), we confirmed the expression of EmGFP in hiPSC-CMs by flow cytometric analysis, which showed most cells were positive for cardiac Troponin T (cTNT; cardiac-specific

marker) (Fig. 1d and e). This cell line (#4-12-20) was shown to have normal karyotype (Fig. 1f) and used in subsequent experiments as the TNNI1-EmGFP hiPSC line. To generate a double reporter line from clone #4-12-20, the same strategy was conducted when inserting the 2A-mCherry cassette into the locus of the TNNI3 stop codon. We also confirmed the insertion, the elimination of selection elements, and normal karyotype in hiPSC clone #8-5-3 (Supplementary Fig. 1c–e). The expression kinetics of EmGFP and mCherry mirrored the expression of TNNI1 and TNNI3, respectively (Supplementary Fig. 1f).

### Molecule screening for cardiac maturation stimulators.
To establish a scalable and reproducible method to screen cardiac maturation, we performed large-scale cardiac differentiation using a bioreactor to form embryoid bodies (EBs). On day 14 of the differentiation, ~90% of hiPSC-derived cells were positive for TNNI1 (Fig. 2a–c). As a positive control for TNNI3 expression, we tested 12 commercially available compounds reported to facilitate cardiac maturation both as single compounds and in combinations[10] (Supplementary Fig. 2a). After the EB dissociation, we found that the combination of endothelin-1 (ET-1) and protein kinase C inhibitor (PKCi) notably increased the mCherry intensity compared to vehicle treatment (Supplementary Fig. 2b and c). We thus used ET-1 and PKCi as the positive control in the subsequent screening. We extensively screened a library of 9048 chemical compounds for the enhancement of TNNI3-mCherry expression (see the "Methods" section and Supplementary Fig. 2d). As a homogenized standard, all compounds were added at 3 μM, resulting in 387 candidates based on the median values plus 3 median absolute deviations (MAD) scores (Fig. 2d). We investigated these compounds for reproducibility, dose-dependency, cell morphology, and mRNA expression for TNNI3, narrowing our candidate list down to eight (Fig. 2e). These eight stimulators increased the mCherry intensity compared to the positive control (Fig. 2f). Immunostaining for one of the candidates, T112, showed that T112-treated cells were strongly positive for mCherry in a dose-dependent manner (Fig. 2g). Interestingly, treatment with any one of five other stimulators (T433, T623, T790, T746, and T220) at 1 μM reduced the cell number, and four of them (T433, T623, T790, and T220) were annotated as inhibitors of cell-cycle-related factors, such as SKP2 (WO2012/002527A1; PCT/JP2011/065148), and of tubulin polymerization (Fig. 2h). This result suggested that arresting the cell cycle might contribute to cardiac maturation. To explore the causal role of cell cycle inhibition in cardiac maturation, we also included T623 as one of the candidates enhancing maturation (Fig. 2f and i). Notably, no significant difference was observed in cell number between the positive control and T112-treated group (Fig. 2h). By using a time-resolved (TR) FRET assay, we observed T112 (2-[2-(4-bromophenyl)-1H-benzimidazol-5-yl]-5-(morpholin-4-yl)-2,3-dihydro-1H-isoindol-1-one) (Supplementary Fig. 2e) to have ERRγ-agonistic activity with stronger effects than the commercially available ERRγ agonist GSK4716 (Supplementary Fig. 2f). Previous reports have shown that ERRγ is essential for the metabolic switch in postnatal heart in a mouse model[14]. Taken together, the observed T112 effect on cardiac maturation in terms of TNNI3 expression may involve distinct signaling pathways compared to the other compounds. In order to conduct a comparative analysis of single compounds and combinations, we selected T112 and T623 for subsequent analysis.

To investigate whether T112 and T623 could stimulate cardiac maturation in a scalable differentiation method, we sought the appropriate timing and concentration of their application to EBs. To that end, we treated EBs at days 6, 8, 10, or 12 until day 16 and analyzed them at day 16 in order to assess changes in

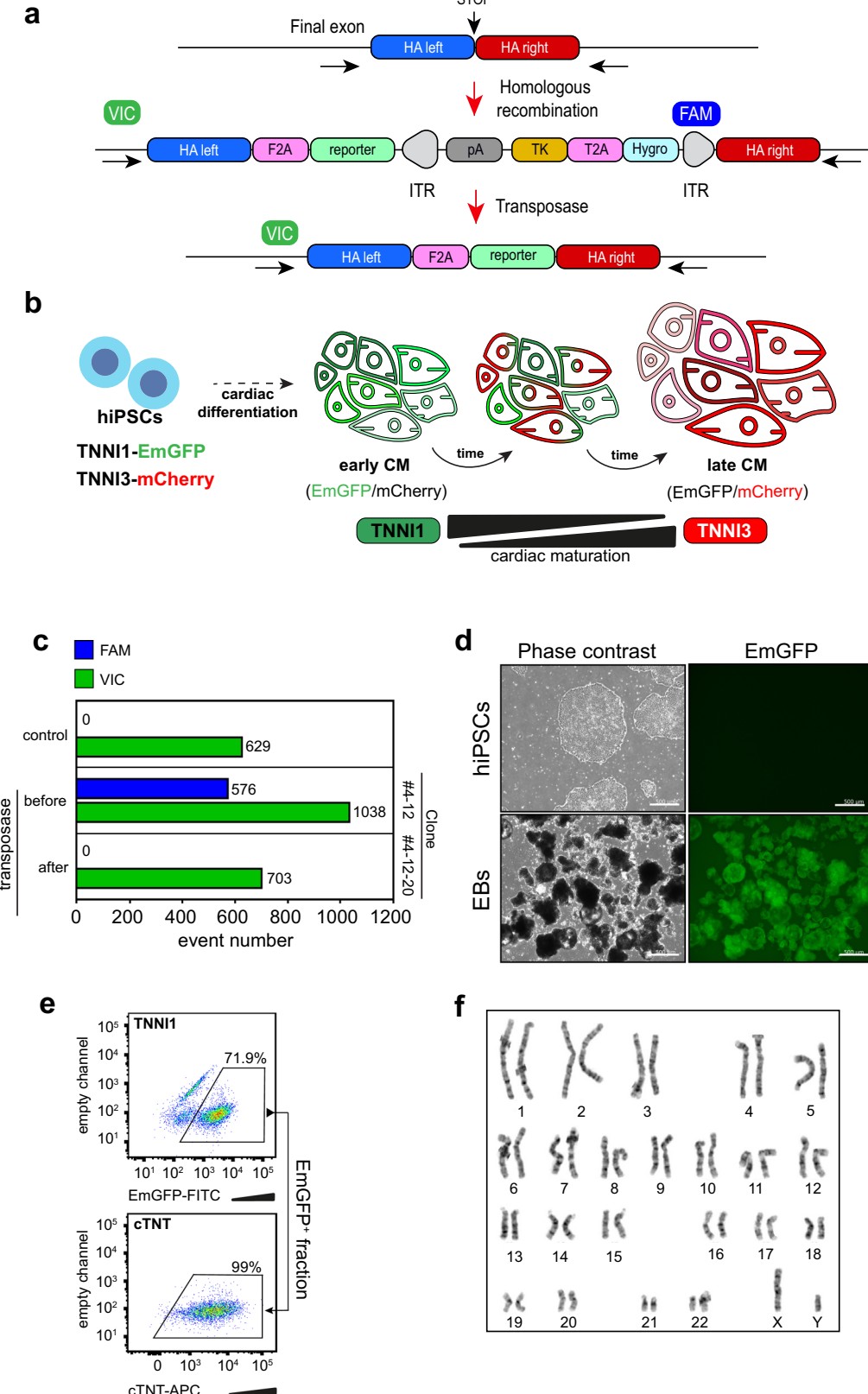

**Fig. 1 Generation of the hiPSC TNNI1^EmGFP and TNNI3^mCherry dual reporter. a** Schematic for the construction of the reporter cell line and the homologous recombination strategy using the CRISPR/Cas9 system. FAM (6-carboxyfluorescein) and VIC (VIC™ fluorescent dye) are fluorescent probes for ddPCR. **b** Schematic representation of the TNNI1^EmGFP and TNNI3^mCherry dual reporter during cardiac differentiation. **c** ddPCR (FAM probe, ■ blue bars; VIC probe, ■ green bars) of TNNI1^EmGFP knock-in clones before and after treatment with piggyBac transposase. **d** Phase-contrast and EmGFP fluorescence images of hiPSCs (#4-12-20) and day-14 EBs. Scale bars: 500 µm. **e** Flow cytometric analysis and sorting on day 14 of the differentiation for EmGFP+ cells (top) and the EmGFP+-derived cTNT+ cell fraction (bottom). **f** Karyotype of the generated TNNI1-EmGFP knockin clone #4-12-20.

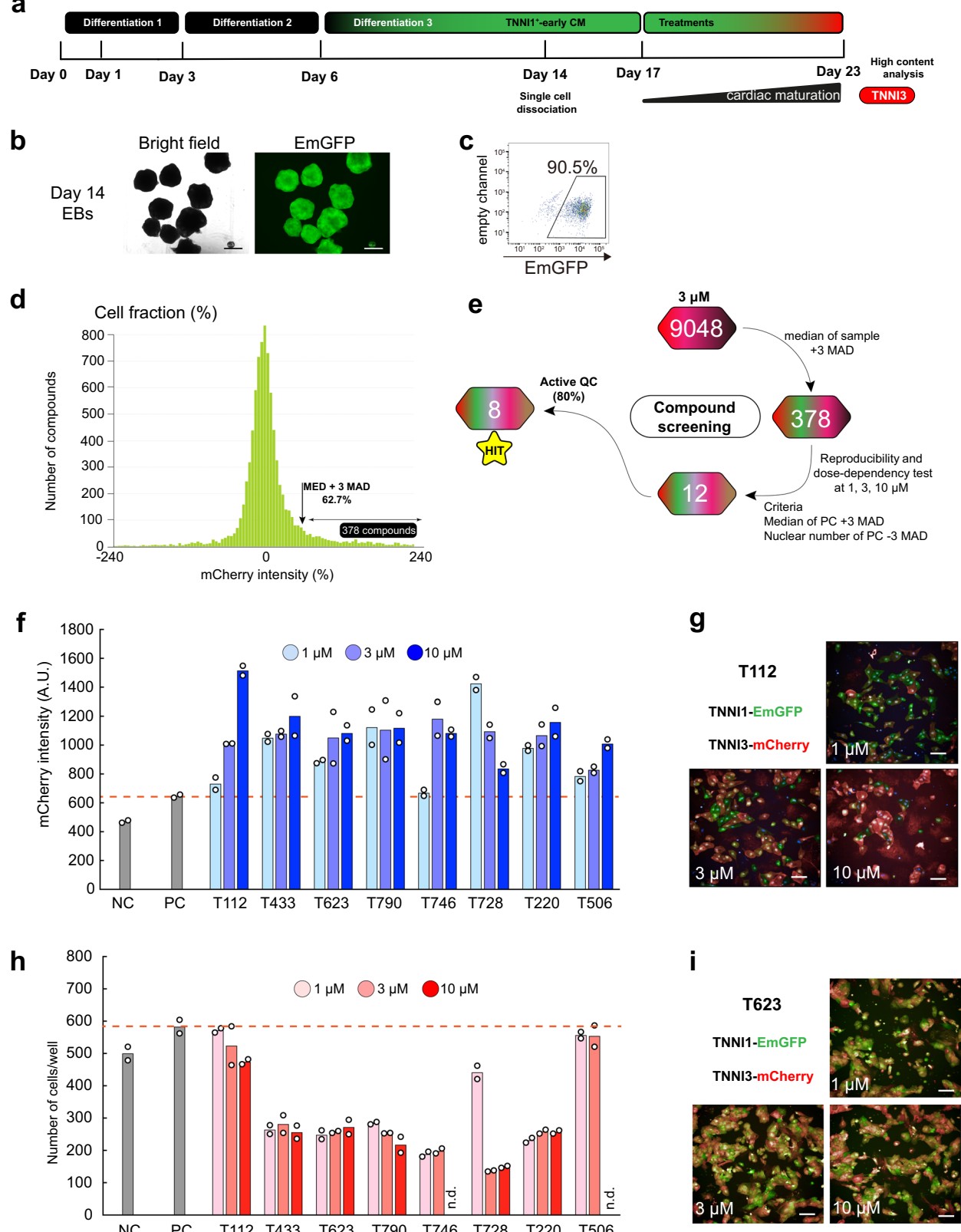

**Fig. 2 Molecular screening for TNNI3^mCherry expression. a** Schematic representation of the cardiac differentiation protocol for cardiac maturation monitoring. **b** Bright field (left) and EmGFP fluorescence (right) images of day-14-derived EBs. Scale bars: 500 μm. **c** Flow cytometric analysis of day-14-derived EBs for TNNI1^EmGFP. **d** Compounds were selected by the median (MED) plus 3 MAD. **e** Screening outline of the chemical compounds. **f** High content imaging for mCherry dose-dependency intensity. $n = 2$ biologically independent samples per group. **g** Dose-dependent T112-derived fluorescence for TNNI1^EmGFP and TNNI3^mCherry. Scale bars: 100 μm. **h** Dose-dependent cell number per well after treatment. n.d. not detected. $n = 2$ biologically independent samples per group. **i** Dose-dependent T623-derived fluorescence for TNNI1^EmGFP and TNNI3^mCherry. Scale bars: 100 μm.

TNNI3-mCherry expression. Treatments with 3 μM T112 from days 8 to 16 or 40 nM T623 from days 10 to 16 resulted in hiPSC-CMs highly expressing TNNI3-mCherry (60.63 ± 3.16% and 42.93 ± 5.29%, respectively) compared with DMSO treatment (10.30 ± 2.16%) (Fig. 3a, b and Supplementary Fig. 3a). In addition, combination treatment of the two (Combo) caused additive TNNI3-mCherry expression (86.53 ± 1.75%) (Fig. 3a, b and Supplementary Fig. 3a), suggesting that the two compounds might exert the mature-derived phenotype through independent mechanisms. Treating the EBs with either higher concentrations or at earlier times caused a notable mCherry increase, but also severe cell damage. We next investigated whether the concomitant treatment of Combo with the positive control could further increase TNNI3-mCherry intensity. However, no significant difference was observed in the mCherry intensity between Combo with or without the positive control (Supplementary Fig. 3b). Thus, we concluded that 3 μM T112 from days 8 to 16 and 40 nM T623 treatment from days 10 to 16 was optimal.

**The transcriptome reveals CM maturation by T112 and T623.** To investigate the transcriptional contribution of T112 and T623, we performed RNA sequencing (RNA-seq) on day-16 hiPSC-CMs treated with DMSO, T623, T112, or Combo. To compare the transcriptomes of the different conditions in relationship to cardiac maturation, a principal-component analysis (PCA) was used. PCA demonstrated a clear separation of DMSO, T623, T112, and Combo maturation transcriptomes with the first two principal components (PCs), accounting for 95.46% of the total variance (Fig. 3c left). A clear separation was also observed for the first and third PCs, accounting for 69.71% of the total variance (Fig. 3c right). A correlation matrix of the samples revealed that a large number of genes was differentially regulated in these four populations (Fig. 3d), with the biggest difference between DMSO and Combo. Similarly, volcano plots visually mapping differentially expressed (DE) genes showed that Combo-treated CM populations mapped the biggest contribution in terms of DE genes distantly from single compound-treated CMs (Fig. 3e), which is consistent with the PCA results.

In order to identify the transcriptomic profiles enabling enhanced cardiac maturation, we first enriched gene sets upregulated by the different treatments by analyzing DE genes in a restrictive run (Fig. 3f). T623-treated CMs showed 344 DE genes, T112-treated CMs showed 947 DE genes, and Combo-treated CMs showed 1254 DE genes, confirming that Combo caused deeper changes in the transcription networks when acquiring maturation hallmarks. Gene ontology (GO) enrichments for the cellular components revealed that T623 enhanced gene sets related to cardiac myofibril and Z-disc; T112 enhanced gene sets related to Z-disc; and Combo enhanced a comprehensive profile of cellular components related to cardiac maturation including: cardiac myofibril and Z-disc as well as mitochondrial respiratory chain I, mitochondrial respiratory chain III, striated muscle thin filament, and integrin complex, thus confirming an advanced maturation phenotype compared with single compound treatments (Fig. 3g). These transcriptional changes highlight individual contributions in structural cardiac maturation and support the benefits of Combo. To strengthen these data, we ran a reactome pathway analysis and found that Combo enriches pathways driving structural and metabolic maturation compared to treatment with T112 or T623 alone (Fig. 4a).

Next, we compared the data with publicly available datasets of murine embryonic stem cell (mESC)-derived cardiac progenitors (mESC-CPs), mESC-derived CMs (mESC-CMs), 20 day-cultured and 1 year-cultured human ESC-CMs (hESC-D20 and hESC-Y1,

respectively), and human adult ventricular CMs[17,18]. We first made density plots of PC2 using public single cell RNA-seq datasets of murine ventricular CMs at E9.5, E14.5, p0, and p21 according to a previous report[4]. According to ortholog gene information, we plotted the maturation levels of hESC-D20, hESC-Y1, and adult ventricular CMs using a published RNA-seq dataset[4]. We then compared the transcriptional data of these three cell types with those of CMs treated with our compounds, finding T112- and T623-treated hiPSC-CMs enhanced cardiac maturation compared with DMSO-treated hiPSC-CMs and approached the maturation of hESC-Y1 (Supplementary Fig. 3c). Moreover, hiPSC-CMs treated with Combo exceeded the maturation of hESC-Y1 (Supplementary Fig. 3c). Nevertheless, the analysis indicated that Combo-induced transcriptional contributions were different and insufficient compared to adult ventricular CMs (Supplementary Fig. 3c).

To explore the contribution of each compound on cardiac-specific genes related to maturation, we then compared genes for the cardiac sarcomere, conduction, and metabolism maturation in our hiPSC-CMs and adult ventricular CMs. The expression patterns of TNNT3, MYH7, SCN5A, and KCNJ2 increased, while those of TNNI1, MYH6, HCN1, and HCN4 decreased in hiPSC-CMs treated with T112, T623, or Combo, showing the cells approached adult ventricular CMs (Fig. 4b). In addition, the expression patterns of glycolysis- and fatty acid oxidation (FAO)-related genes in hiPSC-CMs treated with T112, T623, or Combo also showed a change towards adult ventricular CMs (Supplementary Fig. 3d).

**T112 promotes cardiac maturation in multiple ways.** Next, we investigated the mitochondrial content of hiPSC-CMs treated with T112 and Combo using mitochondrial staining and flow cytometric analysis. We found populations with the highest fluorescence intensity (Fraction 1) increased and populations with middle and low fluorescence intensities (Fractions 2 and 3, respectively) decreased with treatment but without an additive effect (Supplementary Fig. 4a). Furthermore, we performed flux analysis to evaluate the oxygen consumption rate (OCR), an indicator of mitochondrial function. We found that ATP production together with the basal and maximal OCR were significantly higher in hiPSC-CMs treated with T112 or Combo (Fig. 5a and b). To avoid heterogeneity between cell lines, we also investigated the causal role of T112 at accelerating the metabolic maturation of CMs derived from other hiPSC lines (409B2 and 1390C1). hiPSCs were differentiated into CMs and treated with T112, T623, or Combo following the aforementioned protocol. The mitochondrial content was analyzed by mitochondrial staining using CMs detected with miRNA-switch technology[19]. For flux analysis, the CMs were sorted using the cardiac-specific surface markers SIRPA$^+$ LIN$^-$ (CD31, CD49a, CD90, and CD140b)[20]. Again, the flow cytometric analysis showed that Fraction 1 increased and Fractions 2 and 3 decreased in the T112-treated and Combo-treated groups (Supplementary Fig. 4a). In addition, flux analysis demonstrated a higher ATP production and basal and maximal OCR in hiPSC (409B2)-CMs treated with T112 or Combo (Supplementary Fig. 4b).

To examine the cTnI/ssTnI (TNNI3/TNNI1) protein isoform ratio as a quantitative marker of the cardiac maturation status, we collected protein samples treated with DMSO, T112, T623, or Combo and a sample of commercially available adult heart tissue. Western blot analysis showed that cTnI isoform proteins were expressed in hiPSC-CMs treated with T112 or Combo and that the cTnI/ssTnI protein isoform ratio was increased in both groups (Fig. 5c). In addition, increases of the TNNI3 expression and cTnI/ssTnI protein isoform ratio in hiPSC-CMs treated with

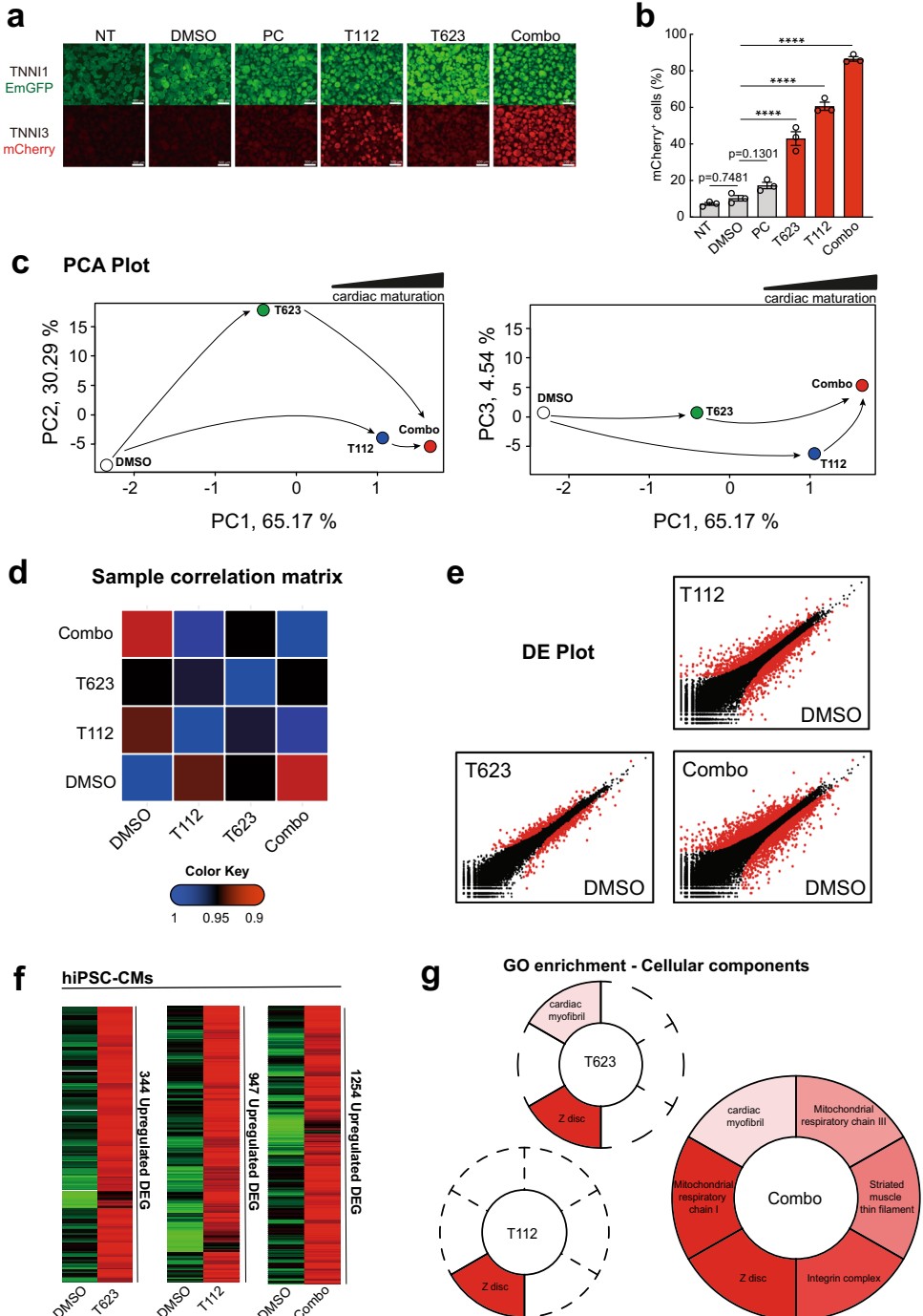

**Fig. 3 T112 and T623 enhanced the cardiac maturation of hiPSC-CMs. a** Fluorescence images of day-16-derived EBs for TNNI1^EmGFP and TNNI3^mCherry expression. Scale bars: 500 μm. NT no treatment, PC positive control. **b** Flow cytometric analysis of day-16-derived cells for TNNI3^mCherry expression. $n = 3$ independent experiments per group. Data are the mean ± SEM; ****$P < 0.0001$ compared to DMSO using one-way ANOVA followed by Dunnett's test. NT no treatment, PC positive control. **c** Principal-component analysis (PCA) in correlation with the maturation of CMs for more than 58,000 genes expressed in the four groups. Data are shown as average PCA for each group. $n = 3$ independent experiments per group. **d** Sample correlation matrix resulting from the consensus clustering analysis of mRNA data and showing transcriptomic differences among treatments. Data are shown as averages for each group. $n = 3$ independent experiments per group. **e** Volcano plots for DE genes in accordance with the cardiac maturation phenotype. **f** Summary of upregulated DE genes in either T623/T112-treated CMs or Combo-treated CMs compared to DMSO. The number of upregulated genes (at least 2-fold) is depicted. Left: 344 unique elements detected in the T623-treated CMs compared to DMSO. Middle: 947 unique elements detected in the T112-treated CMs compared to DMSO. Right: 1254 unique elements detected in the Combo-treated CMs compared to DMSO. **g** Analysis of the top-6 cardiac-related significant GO terms for the cellular components of T623-, T112- and Combo-treated CMs.

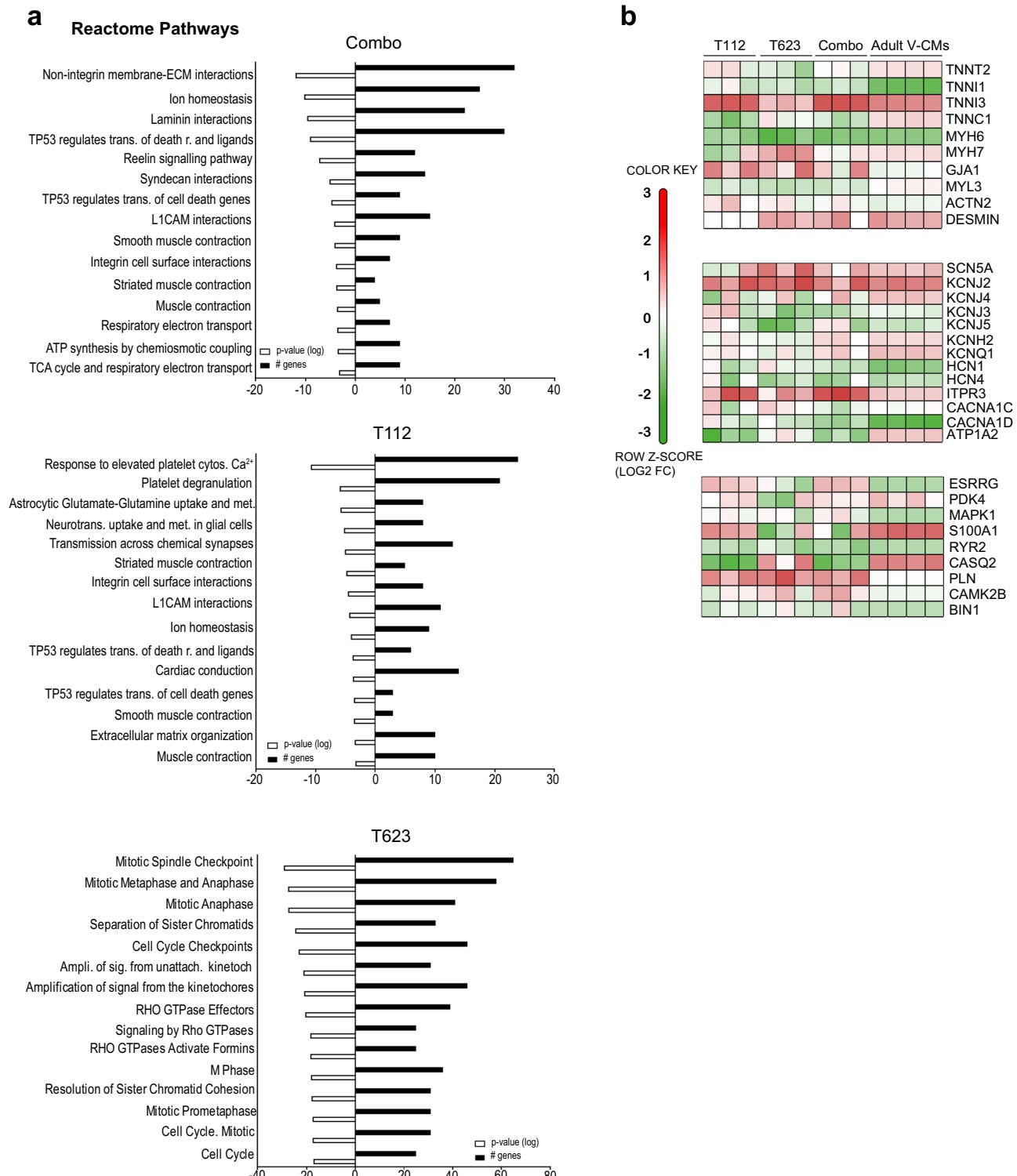

**Fig. 4 Transcriptomic pathways and gene sets enrichment. a** Reactome pathway analysis for T623-, T112- and Combo-treated CMs. Enriched pathways were calculated using the binomial test over DE genes between individual conditions compared to DMSO. The top 15 statistically significant pathways are listed. *P*-values are displayed as logarithmic form (log). **b** Gene set analysis of cardiac maturation-specific genes.

T112 or Combo were observed in other hiPSC lines (409B2 and 1390C1) (Supplementary Fig. 4c and d). However, hiPSC-CMs treated with T112 or Combo expressed ssTnI (TNNI1), which is unlike adult heart. The cTnI/ssTnI protein isoform switch in mouse ventricular heart begins irreversibly during the neonatal period, and the switch eventually becomes 100% cTnI in the adult hearts of all mammalian species including humans[7]. Thus, our data suggested that hiPSC-CMs treated with T112 or Combo had the maturity of neonatal CMs.

To further investigate the cardiac maturation effects induced by T112 and T623 treatment, we measured the cell area, roundness, and sarcomere length. The size of hiPSC-CMs treated with DMSO or T623 was similar and widely distributed (Fig. 5d). The size was larger, and the distribution of the cell area was shifted in

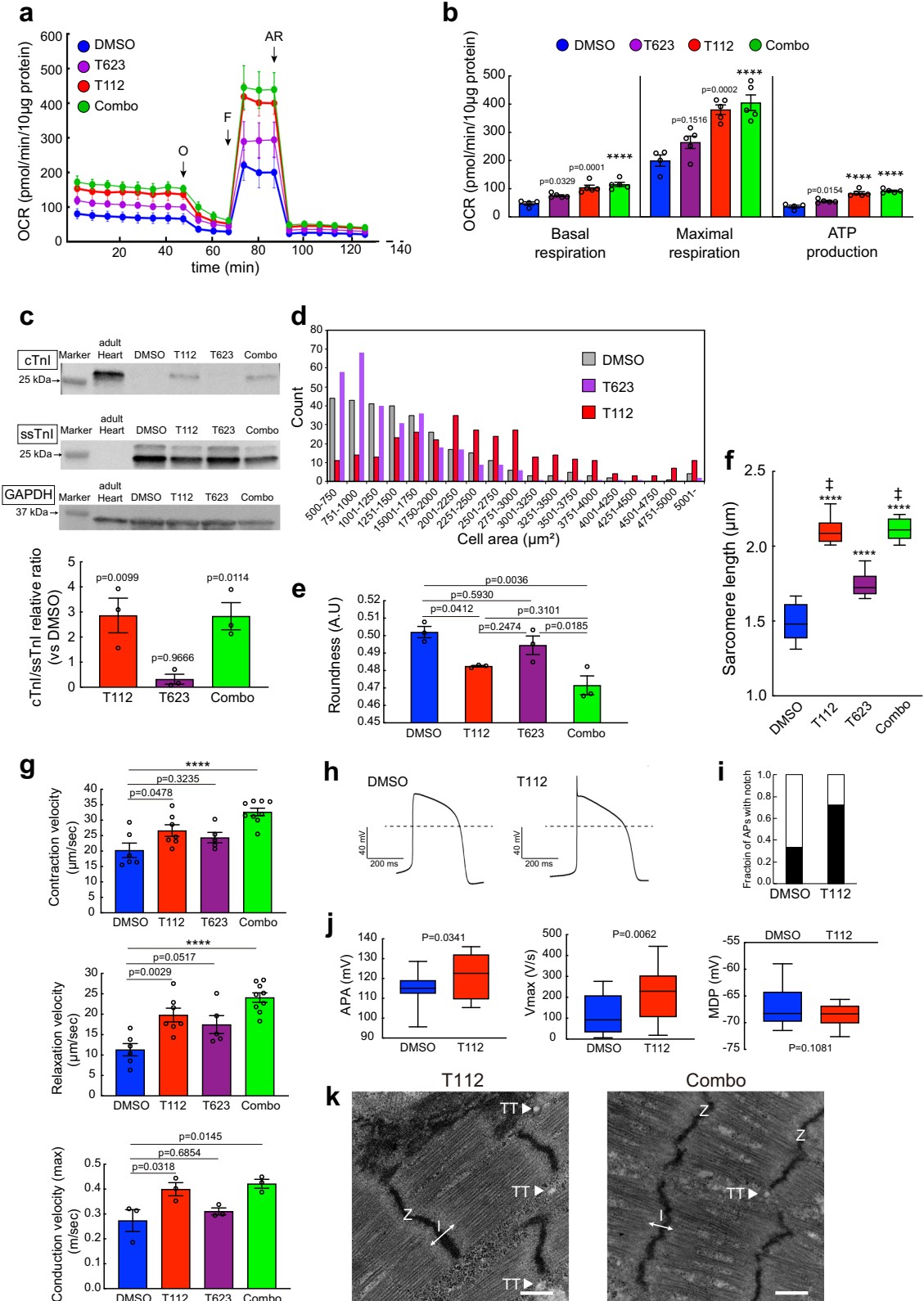

hiPSC-CMs treated with T112 (Fig. 5d and Supplementary Fig. 4e). In addition, the roundness index was significantly decreased (Fig. 5e). Transmission electron microscopy showed that the sarcomere length of hiPSC-CMs treated with T112 or Combo was significantly longer than that of hiPSC-CMs treated with T623, which was significantly longer than that of hiPSC-CMs treated with DMSO (Fig. 5f and Supplementary Fig. 4f).

Since SKP2 inhibitors might ultimately induce cell-cycle arrest, we studied the causal role of T623 on cell cycle activity and cardiac maturation. We found 1 µM reduced the cell proliferation ratio, but a suboptimal concentration of 40 nM did not (Supplementary Fig. 4g). SKP2 inhibitors have apoptotic effects[21]. Consistently, we found cell debris increased with 100 nM T623. These results suggest that T623 enhances the hallmarks of

**Fig. 5 Metabolic and structural functionality of hiPSC-CMs. a** Mitochondrial respiration rates of hiPSC-CMs treated with DMSO ($n = 4$), T112 ($n = 5$), T623 ($n = 5$), or Combo ($n = 5$) (O, *Oligomycin*; F, *FCCP, phenylhydrazone*; AR, *Antimycin A and Rotenone*). $n =$ biologically independent samples per group. Data are the mean ± SEM. **b** Basal and maximal OCR and ATP production of hiPSC-CMs treated with DMSO ($n = 4$), T112 ($n = 5$), T623 ($n = 5$), or Combo ($n = 5$). $n =$ biologically independent samples per group. Data are the mean ± SEM. Statistical analysis was compared to DMSO using one-way ANOVA followed by Dunnett's test. ****$P < 0.0001$. **c** top: Representative western blots of cTnI, ssTnI, and GAPDH proteins in hiPSC-CMs treated with DMSO, T112, T623, or Combo. For ssTnI, two bands were detected: the shorter one is wild type ssTnI and the longer one is ssTnI with F2A peptides. Bottom: cTnI/ssTnI relative ratio. $n = 3$ independent experiments per group. Data are the mean ± SEM. Statistical analysis was compared to DMSO using one-way ANOVA followed by Dunnett's test. **d** Cell area of hiPSC-CMs treated with DMSO, T112, or T623 measured by high content imaging. For each group, $n = 300$ cells over three independent experiments. **e** Roundness of hiPSC-CMs treated with DMSO, T112, T623, or Combo. Each group, $n = 3$ independent experiments. Data are the mean ± SEM. Statistical analysis was done using one-way ANOVA followed by Tukey's HSD test. **f** Sarcomere length in hiPSC-CMs treated with DMSO, T112, T623, or Combo. Boxes represent 25th–75th percentiles; whiskers represent the minimum and maximum ranges; horizontal lines indicate the median values. $n = 15$, one measurement per TEM image from 15 TEM images over two consecutive experiments; ****$P < 0.0001$ compared to DMSO, ‡$P < 0.0001$ compared to T623 using one-way ANOVA followed by Tukey's HSD test. **g** Contractile properties of hiPSC-CMs treated with DMSO ($n = 6$), T112 ($n = 7$), T623 ($n = 5$), or Combo ($n = 9$). Top: Contraction velocity. Middle: Relaxation velocity. $n =$ biologically independent samples examined over three independent experiments. bottom: Maximum conduction velocity. $n = 3$ independent experiments. Statistical analysis was compared to DMSO using one-way ANOVA followed by Dunnett's test. ****$P < 0.0001$. **h** Representative AP recorded from hiPSC-CMs treated with DMSO or T112. **i** The fraction of APs with a notch (black). **j** Electrophysiology data of APA, Vmax and MDP in hiPSC-CMs treated with DMSO ($n = 21$) or T112 ($n = 18$) over 3 independent experiments. Boxes represent 25th–75th percentiles; whiskers represent the minimum and maximum ranges; horizontal lines indicate the median values. Statistical analysis was done using unpaired two-tailed *t*-test. **k** Typical high magnification TEM images of hiPSC-CMs treated with T112 or Combo. Scale bars: 500 nm. I: I-band; TT: T-tubule; Z: Z line. All results are from #8-5-3.

maturation in a dose-dependent manner without affecting the cell cycle status. Counteracting this effect, treating EBs with 3 μM T112 decreased the number of Ki-67⁺ cells and increased the number of Ki-67⁻/Hoechst⁺ cells (i.e., binucleate cells; Supplementary Fig. 4h).

We next assessed whether T112, T623, or Combo enhances the contractile properties and electrical connectivity (conduction velocity) of hiPSC-CMs. Motion vector analysis data showed that the contraction, relaxation, and maximum conduction velocities were significantly increased in hiPSC-CMs treated with T112 or Combo (Fig. 5g). These effects on the relaxation and maximum conduction velocities were also observed in another hiPSC line (Supplementary Fig. 4i).

To determine whether T112 enhances the electrical maturation of hiPSC-CMs, we measured action potentials (APs) using patch-clamp electrophysiology (Fig. 5h). The number of APs with a typically mature notch was increased in hiPSC-CMs treated with T112 (Fig. 5i and Supplementary Fig. 4j). AP amplitudes (APA) and maximum upstroke velocity ($V_{max}$) in hiPSC-CMs treated with T112 were significantly increased compared with hiPSC-CMs treated with DMSO (Fig. 5j). On the other hand, maximum diastolic potential (MDP) showed no significant difference between the two groups (Fig. 5j).

Transverse tubule (T-tubule) formation is one of the most important indicators of maturation in terms of excitation–contraction coupling and remains a major hurdle in the maturation of hiPSC-CMs. We investigated whether T112, T623, or Combo enhances T-tubule formation. Transmission electron microscopy data revealed T-tubule formation in hiPSC-CMs treated with T112 or Combo in multiple hiPSC lines (Fig. 5k and Supplementary Fig. 5f). In addition, hiPSC-CMs displayed clearly aligned sarcomeres with I-band, M-line, and Z lines (Supplementary Fig. 5a–f). On the other hand, neither T-tubule formation nor the M-line was detectable in hiPSC-CMs treated with DMSO or T623.

Collectively, these data confirm that T112 enhanced the metabolic and sarcomere maturation and the contractile and electrophysiological properties of hiPSC-CMs along with causing an organized ultrastructure and the presence of T-tubules.

**ERRγ plays a causal role in regulating cardiac maturation.** To investigate the functional role of ERRγ-derived TNNI3 expression, we performed loss-of-function studies. To that end, we

generated an ERRγ knockout (ERRγKO) hiPSC line from clone #8-5-3 using CRISPR/Cas9. We inserted the stop sequence in the middle of exon 2 encoding the DNA-binding domain, a common region for all ERRγ variants (Supplementary Fig. 6a)[14]. We confirmed the loss of ERRγ expression in ERRγKO hiPSC-CMs by western blot analysis and determined a single hiPSC clone (#1-6-11) as an ERRγKO hiPSC line (Supplementary Fig. 6b and c). No significant difference existed in the cardiac differentiation efficiency of ERRγKO and wild-type hiPSCs, but ERRγKO hiPSC-CMs treated with 3 μM T112 failed to upregulate TNNI3-mCherry expression (14.45 ± 2.98%) (Fig. 6a and b). In contrast, ERRγKO hiPSC-CMs and wild-type hiPSC-CMs responded equally to 40 nM T623 (Fig. 6a and b), confirming a causal role of ERRγ in cardiac maturation in terms of TNNI3 expression. We also performed a metabolic flux analysis to evaluate the loss of metabolic maturation controlled by T112 treatment. We assessed OCR in ERRγKO hiPSC-CMs treated with DMSO or T112 and found that ERRγKO hiPSC-CMs treated with T112 did not increase their ATP production or their basal and maximal OCR (Fig. 6c and d), confirming the role of ERRγ in metabolic maturation. Altogether, our data demonstrated that T112 contributes to CM maturation through ERRγ signaling (Fig. 7).

## Discussion

Regenerative therapies using engineered CMs and tissue models have benefited from growing knowledge of the molecular and cellular bases of cardiac differentiation and maturation. hiPSCs represent a major source of engineered differentiated CMs, but current hiPSC-CMs exhibit poor maturity. Here, we identified two candidate compounds that accelerate the maturation of hiPSC-CMs via a high-throughput screening that detected the upregulation of TNNI3 expression. Notably, an ERRγ agonist induced sarcomere and metabolic maturation, enhanced contractile and electrical properties, and promoted the generation of T-tubules without any mechanical constraints or electrical stimuli.

In mouse, ERRγ is known to act as a key regulator of the metabolic switch to oxidative metabolism[14]. Alaynick et al. revealed that ERRγKO mice have lactatemia and electrophysiological abnormalities, such as depolarization (e.g. increase in the duration of the QRS complex) and repolarization (e.g. increase in the durations of the ST, QT, and QTc intervals), driven by disruption of the perinatal metabolic switch, which results in death within a week after birth. In addition,

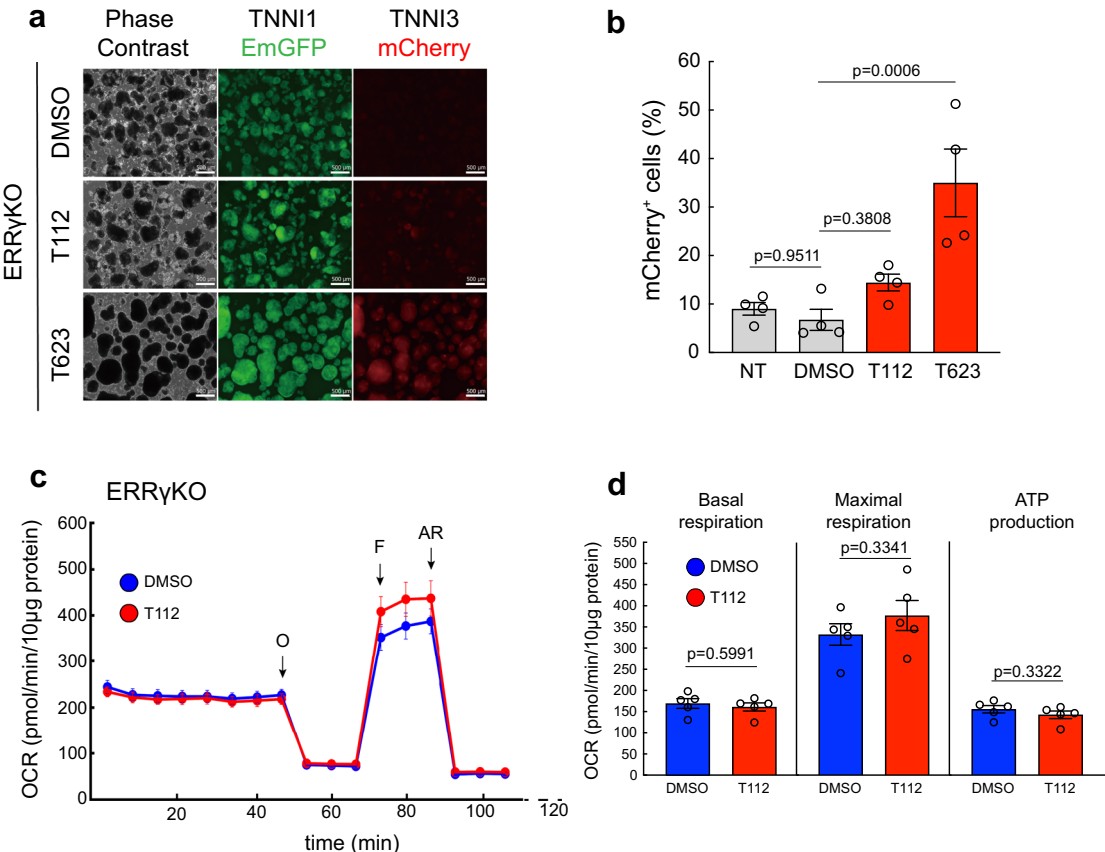

**Fig. 6 Stage-specific causal role of ERRγ in regulating TNNI3-driven cardiac maturation. a** Fluorescence images of ERRγKO hiPSC-CMs treated with DMSO, T112, or T623 for TNNI1[EmGFP] and TNNI3[mCherry] expression. Scale bars: 500 μm. **b** Flow cytometric analysis of day-16 CMs derived from ERRγKO hiPSCs for TNNI3[mCherry] expression. $n = 4$ independent experiments per group. Data are the mean ± SEM; Statistical analysis was compared to DMSO using one-way ANOVA followed by Dunnett's test. NT no treatment. **c** Mitochondrial respiration rates of ERRγKO hiPSC-CMs treated with either DMSO or T112 (O *Oligomycin*; F FCCP, *phenylhydrazone*; AR *Antimycin A and Rotenone*). $n = 5$ biologically independent samples per group. **d** Basal and maximal OCR and ATP production of ERRγKO hiPSC-CMs treated with either DMSO or T112. Statistical analysis was done using unpaired two-tailed *t*-tests. $n = 5$ biologically independent samples per group. Data are the mean ± SEM; Statistical analysis was using unpaired two-tailed *t*-test.

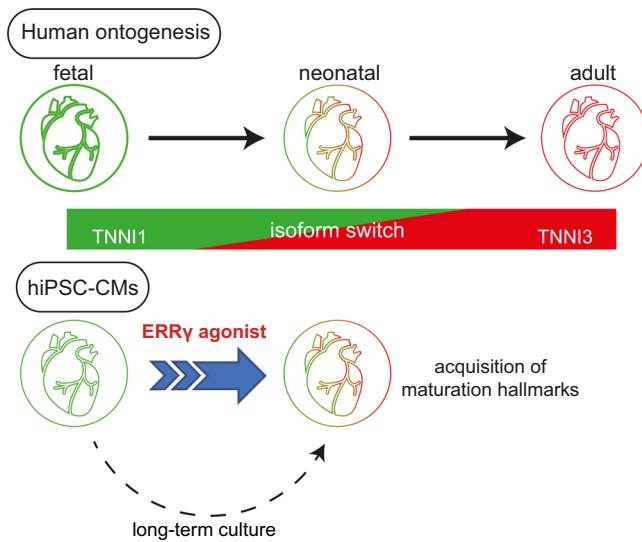

**Fig. 7 hiPSC cardiac maturation model.** Schematic representation of an ERRγ agonist inducing the cardiac neonatal-like phenotype.

Sakamoto et al. demonstrated that ERR signaling is essential for postnatal mitochondrial maturation and that postnatal ERRα/ERRγ knockdown mice develop cardiomyopathy with a lack of mitochondrial maturation[22]. They also showed ERRα/ERRγ-deficient hiPSC-CMs have mitochondrial dysfunction, contractile dysfunction and abnormal excitation–contraction coupling compared with wild-type hiPSC-CMs. Yoshihara et al. showed that the forced expression of ERRγ in hiPSC-derived β-like (iβ) cells enhanced maturation phenotypes, such as c-peptide secretion and expression profiles, and iβ cells with forced ERRγ expression restored glucose homeostasis in β cell-deficient mice[23]. In the present study, we found an ERRγ agonist (T112) from our screening panel, which has a different chemical structure from the commercially available ERRγ agonist (GSK4716), and revealed that it increased mitochondria mass and function in hiPSC-CMs as well as TNNI3 upregulation (Fig. 5a and b and Supplementary Fig. 4a and b). In contrast, GSK4716 failed to increase TNNI3 expression (Supplementary Fig. 6d). A LanthaScreen TR-FRET coactivator assay showed that GSK4176 has weaker activity than T112 (Supplementary Fig. 2f). These results indicate that ERRγ agonistic activity can enhance the cardiac maturation of hiPSC-CMs in a dose-dependent manner with a threshold defined by the effective concentration inducing TNNI3 expression. In addition,

T112 has potential applications to other cell types and for the in vitro study of ERRγ.

ERRs, including ERRα, ERRβ, and ERRγ, are orphan nuclear hormone receptors contributing to physiological signaling pathways in the heart, pancreatic β-cells, liver, and skeletal muscle. Especially in the heart, several studies have revealed that ERRs regulate genes encoding FAO and oxidative phosphorylation (OXPHOS)[14,24,25]. The overexpression of ERRγ in mouse heart induces hypertrophic phenotypes by directly binding to GATA4 promoter. Consistently, the knockdown of ERRγ, the application of an inverse agonist of ERRγ or the inhibition of GATA4 blocks cardiac hypertrophy[26]. In our study, RNA-seq data showed that the expression of FAO-related genes (FABP3, ACADM, ACADM, HADHA, CPT2), OXPHOS-related genes (NDUFB3, NDUFC2, NDUFB8, NDUFS7, COX5B, NDUFB7, NDUFA3, TFAM, MTIF2, IDH1, COX15) and citric acid cycle-related genes (MDH1, IDH2, and DLST) changed with T112 treatment (Supplementary Fig. 3d), supporting previous studies in the field. T112 treatment also increased the mitochondria content (Supplementary Fig. 4a). These data suggest that T112 induces mitochondrial biogenesis and maturation in hiPSC-CMs through ERRγ activity. In addition, we showed that T112 increases the cell area, sarcomere length, cTnI/ssTnI ratio, and contractile properties, decreases cellular roundness, generates detectable M-lines, and induces T-tubule formation in multiple hiPSC lines (Fig. 5c–g and i and Supplementary Fig. 4d–f, i and 5), demonstrating that T112 enhances not only TNNI3 expression but also sarcomere maturation through ERRγ activity. Our results represent an important advance, because no method for generating hiPSC-CMs with T-tubule formation in a simple protocol using a conventional culture environment with a single compound thus far has been reported.

We also found that T623, an SKP2 inhibitor, induces primary maturation in terms of early TNNI3 expression. Treatment with 40 nM T623 in EBs notably increased mCherry-positive cells and promoted the expression of maturation-related genes (Fig. 3a–g and Supplementary Fig. 3,a, c and d). In addition, T623 treatment increased the sarcomere length compared with DMSO treatment (Fig. 5f and Supplementary Fig. 4f). However, T623 treatment did not increase the mitochondrial activity or cell area or generate T-tubules. The reason is possibly attributable to the different concentrations required for the upregulation of TNNI3 and change of other maturation-related parameters. Although T623 can terminally reduce cell-cycle active cells in a dose-dependent manner (Supplementary Fig. 4g and h), higher concentrations do not lead to a more mature phenotype, suggesting that T623-induced maturation is not mediated by cell cycle arrest.

Our RNA-seq-based transcriptomics analysis also revealed that even though hiPSC-CMs can mature upon a single-compound treatment, a combination of compounds enhances a more comprehensive gene set for cardiac maturation, particularly for cellular components involved in structural and metabolic maturation (Fig. 3g). In addition, the electrophysiological properties of the compound-treated groups were similar to that of adult ventricular CMs (Fig. 4b). Notably, treatment with T112 significantly increased the conduction velocity (Fig. 4g and Supplementary Fig. 4i), indicating enhanced electrical connectivity. Furthermore, patch clamp data (Fig. 5h) showed that T112 significantly enhances the APA and $V_{max}$ in hiPSC-CMs, which is consistent with our RNA-seq data.

To date, several studies reported diverse methodologies to induce one or more hallmarks of cardiac maturation. Ronaldson-Bouchard et al. reported an advanced cardiac tissue model using a combination of hiPSC-CMs and human dermal fibroblasts that underwent electrical stimulation-based intensity training[27]. Tiburcy et al. engineered human myocardium composed of hPSC-CMs and human foreskin fibroblasts that showed structural and functional properties similar to postnatal myocardium[28]. Recently, Giacomelli et al. showed that 3D microtissues consisting of hiPSC-CMs, -cardiac fibroblasts and -endothelial cells have enhanced cardiac maturation and that dibutyryl cAMP (dbcAMP) induces electrophysiological maturation[29]. Previous studies reported that cAMP regulates ERR expression[30,31]. Based on these findings, we assessed whether cAMP induces CM maturation via an ERRγ-mediated pathway by investigating the maturation-inducing effect of cAMP on ERRγKO hiPSC-CMs. We found the addition of dbcAMP did not increase the expression of TNNI3-mCherry in ERRγKO hiPSC-CMs, suggesting that cardiac maturation through cAMP is mediated by ERRγ (Supplementary Fig. 6e). To elucidate the regulatory mechanisms of the maturation of hiPSC-CMs, further investigation is required.

3D tissues may induce CM maturation by changing the cardiac microenvironment. On the contrary, in the present study, we demonstrated that stimulating intrinsic pathways by chemical compounds can induce CM maturation without electrical stimulation or mechanical stress, which should benefit large-scale 2D-based assays.

Altogether, our results indicate that ERRγ activation plays an important function in the fundamental biology of cardiac maturation. They also have opened new questions regarding the mechanistic and molecular signaling behind ERRγ downstream effectors. However, cardiac maturation during cardiogenesis is probably not the outcome of one single signaling stimulus but exerted through many pathways at multiple cellular functional levels. Nonetheless, our study depicts a major advancement in the field by providing a short and scalable method that generates in vitro hiPSC-CMs displaying sufficient mature features to reliably model diseases and undergo drug screening.

## Methods

**hiPSC culture and small- and large-scale cardiac differentiation.** The hiPSC lines used (1390D4, 1390C1 and 409B2) were established at CiRA. 1390D4 and 1390C1 were cultured on an iMatrix-511 (Nippi)-coated dish in AK02N medium (Ajinomoto) as previously described[32]. 409B2 was cultured on SL10 feeder cells (REPROCELL) in primate ES cell medium (REPROCELL) with human recombinant basic fibroblast growth factor (bFGF) (Wako).

For small-scale cardiac differentiation using a six-well ultra-low attachment plate (Corning), we modified a previously described protocol[19,20]. In brief, hiPSCs were dissociated into single cells using 0.5 × TrypLE select (Thermo Fisher Scientific) (1×TrypLE select diluted with 0.5 mM EDTA) and then suspended in 1.5 mL/well StemPro-34 (Invitrogen) medium supplemented with 2 mM L-glutamine (Invitrogen), $4 \times 10^{-4}$ M monothioglycerol (MTG; Sigma), 50 μg/mL ascorbic acid (AA; Sigma), 150 μg/mL transferrin (Wako), 10 μM ROCK inhibitor (Y-27632; Wako), 0.5% Matrigel (Corning), and 2 ng/mL BMP4 (R&D Systems) at $2 \times 10^6$ cells/well to form EBs. On day 1, 1.5 mL of StemPro-34 medium supplemented with 2 mM L-glutamine, $4 \times 10^{-4}$ M MTG, 50 μg/mL AA, 150 μg/mL transferrin, 10 ng/mL bFGF (final 5 ng/mL), 12 ng/mL activin A (final 5 ng/mL), and 18 ng/mL BMP4 (final 10 ng/mL) was added into the well. On day 3, the EBs were washed with Iscove's modified Dulbecco's media (IMDM; Invitrogen) once and then cultured in 3 mL of StemPro-34 medium containing 2 mM L-glutamine, $4 \times 10^{-4}$ M MTG, 50 μg/mL AA, 150 μg/mL transferrin, 10 ng/mL vascular endothelial growth factor (VEGF; R&D Systems), 1 μM IWP-3 (Stemgent), 0.6 μM dorsomorphin, and 5.4 μM SB431542. On day 6, the medium was exchanged to 2 mL of StemPro-34 medium supplemented with 2 mM L-glutamine, $4 \times 10^{-4}$ M MTG, 50 μg/mL AA, 150 μg/mL transferrin, and 5 ng/mL VEGF. After that, the medium was changed to the same composition medium every 2–3 days. The plate was placed in a hypoxia environment (5% $O_2$) for the first 10 days and then transferred to a normoxia environment.

For large-scale cardiac differentiation using a 30 mL bioreactor (AIBLE), $1 \times 10^7$ dissociated hiPSCs were suspended in 30 mL AK02N without C solution medium supplemented with 2 mM L-glutamine, $4 \times 10^{-4}$ M MTG, 50 μg/mL AA, 150 μg/mL transferrin, 10 μM Y-27632, 0.5% matrigel, and 2 ng/mL BMP4 at 55 rpm to form EBs. On day 1, 9 μL of 10 μg/mL Activin A (final 3 ng/mL), 15 μL of 10 μg/mL bFGF (final 5 ng/mL) and 24 μL of 10 μg/mL BMP4 (final 10 ng/mL) were added into the bioreactor. On day 3, the EBs were collected in a 50 mL tube, washed with IMDM, and then cultured in 30 mL AK02N without C solution medium supplemented with 2 mM L-glutamine, $4 \times 10^{-4}$ M MTG, 50 μg/mL AA, 150 μg/mL transferrin, 10 ng/mL VEGF, 1 μM IWP-3, 0.6 μM dorsomorphin, and 5.4 μM SB431542 in the same bioreactor at 55 rpm. On day 6, 80–90% of the medium was

aspirated, and 25 mL AK02N without C solution medium supplemented with 2 mM L-glutamine, $4 \times 10^{-4}$ M MTG, 50 µg/mL AA, 150 µg/mL transferrin, and 5 ng/mL VEGF was added to the bioreactor. After that, the medium was changed as above every 2–3 days. The bioreactor was placed in a hypoxia environment (5% $O_2$) for the first 10 days and then transferred to a normoxia environment.

**Guide RNA (gRNA) and targeting vector constructions.** Specific gRNAs (AS194 and S254) against TNNI1 were designed for the stop codon of TNNI1 and 26 nucleotides downstream of the stop codon. These gRNAs were subjected to EmGFP integration at the C-terminus of TNNI1 together with a Streptococcus pyogenes Cas9 (SpCas9) mutant, D10A (Supplementary Table 1)[33]. For the genetic modification, the gRNA vectors were constructed on a pENTR/U6 linearized vector in a BLOCK-iT™ U6 RNAi Entry Vector Kit (Thermo Fischer Scientific) according to the manufacturer's standard protocol. The donor templates were prepared from six fragments: the C-terminus sequence of TNNI1 as the left homology arm (1.1 kbp length), a self-cleaving peptide motif from foot-and-mouth disease virus (F2A)[34], EmGFP sequence, the hygromycin resistant gene flanked with piggyBac inverted terminal repeat (ITR) sequences as a selection marker, the 3′-NCR sequence of TNNI1 as the right homology arm (1 kbp length), and a plasmid backbone. The homology arms and EmGFP fragment were prepared by PCR amplification using specific primers that contain overlapped sequences with adjacent fragments at the 5′ position for Gibson assembly (New England BioLabs, MA). F2A peptide sequences were added between the left homology arm and EmGFP fragments as part of the PCR primers. The other fragments were prepared using the restriction enzyme digestion of BsiWI/NsiI and AscI/NotI with the piggyBac multi vector, MV-PGK-Hygro-TK (Transposagen Biopharmaceuticals). After Gibson assembly, the TNNI1 C-terminus, F2A motif and EmGFP sequence were seamlessly and in-frame assembled.

Specific gRNA spanning on the stop codon of TNNI3, S4, was designed. S4 has NGA PAM sequences recognized by SpCas9 mutants D1135V, R1335Q and T1337R (Supplementary Table 1)[35]. The gRNA vector and the donor template, which consisted of the C-terminus sequence of TNNI3 (900 bp length) and the 3′ NCR of TNNI3 (900 bp length), F2A peptide sequences, mCherry sequences, the hygromycin-resistant gene flanked with piggyBac ITR and the backbone, were prepared as described above. In addition, the short donor template (Fig. 1a) was assembled from three fragments: the C-terminus sequence of TNNI3 (900 bp length) with a silent mutation, the 3′-NCR sequence of TNNI3 (900 bp length) and the plasmid backbone.

For ERRγKO cells, specific gRNAs targeting exon2, which is the common region for all variants of ERRγ, were selected (AS345 and S393) (Supplementary Table 1). The donor templates were prepared from four fragments: the upstream sequences of AS345 (897 bp length), the hygromycin-resistant gene flanked with piggyBac ITR, the downstream sequences of S393 (808 bp length) and the plasmid backbone. The translation of ERRγ was designed to terminate after the left homology arm because of the appearance of an artificial stop codon (TAA) on the edge of the ITR sequence.

**Generation of knockin (KI) and KO hiPSC lines.** To generate KI and KO lines, we first dissociated hiPSCs (1390D4) into single cells using 0.5× TrypLE Select and a cell scraper (IWAKI), and then electroporated the vector complex into $2 \times 10^6$ cells using the Neon® Transfection System (Thermo Fisher Scientific) (100 µL Kit, 1200 V, 20 ms, 2 pulses). Cells were seeded on two iMatrix-511-coated 6 cm dishes ($1 \times 10^6$ cells/dish) in AK02N medium supplemented with 10 µM Y-27632 for 1 day. On the next day, we changed the medium to AK02N medium without Y-27632. On day 3, we changed the medium to AK02N medium supplemented with 50 µg/mL hygromycin to select positive clones for 3 days. On days 10–14, we picked the survived colonies and expanded and analyzed them.

To eliminate the selection elements, we electroporated piggyBac Transposase mRNA (Transposgen) into $1 \times 10^5$ hiPSCs using the Neon® Transfection System (10 µL Kit, 1200 V, 20 ms, 2 pulses) and seeded the cells on a iMatrix-511-coated 6 cm dish in AK02N medium supplemented with 10 µM Y-27632 for 1 day. On the next day, we changed the medium to AK02N medium without Y-27632. On day 3, we changed the medium to AK02N medium supplemented with 200 nM 1-(2′-deoxy-2′-fluoro-β-D-arabinofuranosyl)-5-iodouracil (FIAU) for 4 days. On days 10–14, we picked the survived colonies and expanded and analyzed them.

**High throughput screening.** Day-14 EBs differentiated using the bioreactor were treated with 2 mg/mL collagenase type I for 2 h and then treated with 0.25% trypsin/EDTA for 10 min. To neutralize trypsin, 50% FBS/IMDM was added to the EBs, and we gently pipetted the EBs to dissociate them into single cells. The cells were centrifuged at $200 \times g$ for 5 min and suspended in AK02N without C solution medium supplemented with 2 mM L-glutamine, $4 \times 10^{-4}$ M MTG, 50 µg/mL AA, 150 µg/mL transferrin, and 5 ng/mL VEGF. The cells were seeded in CellCarrier-384 Ultra Microplates (PerkinElmer) covered with iMatrix-511 at a density of 10,000 cells/well by an automated dispenser (Multidrop Combi; Thermo Fisher Scientific). The cells were then incubated at 37 °C and 5% $CO_2$ to allow them to attach.

We extensively screened a library of 9048 chemical compounds that enhance TNNI3-mCherry expression. The compounds were screened at 3 µM as a first

screening step. The library was designed to cover a diverse chemical space and included elements from a biologically annotated compound library, focused kinase library and internal/external repositioning library at Takeda Pharmaceutical Company Ltd. The criteria for the compound selection were in vitro pharmacological activity on individual target proteins with an $EC_{50}$ or $IC_{50}$ value ≤1 µM. The activity was determined by cell-free and cell-based binding and functional assays, as shown in Supplementary Fig. 2d. The resultant compound library interacted with 1804 unique proteins. Furthermore, 70% of these target proteins were annotated by multiple compounds, which must be taken into account in order to avoid misinterpretation of the experimental results due to possible off-target effects by the small molecules. After the initial refinement, close analogs of an in-house extensive compound library were tested by an hiPSC-based assay to validate any identifiable structure–activity relationship (SAR). The compounds were added using a multistage workstation (EDR-SX, BioTec, Tokyo, Japan) automatically from a 10 mM stock solution that was prediluted in medium.

For the immunocytostaining, cells were fixed with 4% paraformaldehyde for 15 min. With a plate washer (TECAN HydroSpeed; Tecan Group Ltd., Switzerland), the cells underwent three washing cycles. Then the cells were incubated in 30 µL of permeabilization solution (0.1% Triton X-100 in PBS) for 15 min. After 60 min of pre-incubation with blocking solution (10% goat serum and 0.1% Triton X-100 in PBS), the cells were incubated overnight at 4 °C with rat anti-mCherry antibody (Thermo Fisher Scientific, M11217, 1:200 dilution) with blocking solution. The cells were washed with PBS and then incubated with goat anti-rat antibody conjugated with Alexa Fluor 647 (Thermo Fisher Scientific, A-21247, 1:200 dilution) with blocking solution for 30 min at room temperature. Hoechst (Dojindo Laboratories, 346-07951, 1:5000 dilution) was added in PBS and incubated for 5 min. The cells were washed three times with PBS and stored at 4 °C.

**High-content imaging and image analysis.** During the screening, images were obtained using an OperaPhenix imager (Perkin Elmer) after immunostaining. One image was acquired from each well using a ×10 air NA0.3 objective. The Hoechst fluorescence intensity was detected for the identification of the nuclei area. The cytoplasm area was defined based on the fluorescence intensity of EmGFP. The fluorescence intensity of the Alexa Fluor 647 was quantified to measure reporter protein levels in the defined cytoplasm region. Data for each treatment condition are expressed as the mean and standard deviation of the wells, which were derived from the average of individual cells in each well.

**Droplet digital PCR (ddPCR).** ddPCR (Bio-Rad) was performed according to the manufacturer's instructions. The Taqman probes used are shown in Supplementary Table 2.

**DNA and RNA extraction.** Cells were lysed using a DNeasy Blood & Tissue Kit (QIAGEN) for DNA extraction and a miRNeasy Mini Kit (QIAGEN) for RNA extraction.

**Flow cytometric analysis and cell sorting.** Differentiated EBs were dissociated the same way as in the high throughput screening and suspended in 2% FBS/PBS. We analyzed and sorted the CMs using FACSAria Fusion (BD Biosciences), FACSDiva software (BD Biosciences) and FlowJo software. For Ki-67 staining, we used purified mouse anti-Ki-67 Clone B56 (RUO) (BD Biosciences, 556003, 1:200 dilution) and APC goat anti-mouse IgG (minimal x-reactivity) (Biolegend, 405308, 1:500 dilution). To sort the CMs derived from 1390C1 or 409B2, we used PE/Cyanine7 anti-human CD172a/b (SIRPα/β) antibody (BioLegend, 323808, 1:500 dilution), APC mouse anti-human CD90 (BD, 559869, 1:1000 dilution), APC anti-human CD31 antibody (BioLegend, 303116, 1:500 dilution), Alexa Fluor® 647 anti-human CD49a antibody (BioLegend, 328310, 1:500 dilution) and APC anti-mouse CD140b antibody (BioLegend, 136008, 1:500 dilution).

**Western blots.** Proteins were extracted using RIPA Buffer (Wako) according to the manufacturer's recommendation, and protein concentrations were measured using the Pierce BCA Protein Assay Kit (Thermo Fisher Scientific). Proteins were run by SDS–PAGE using Criterion TGX Precast Gel (Bio-Rad) and transferred to PVDF membranes (Bio-Rad). The primary antibodies used were anti-ERRγ polyclonal antibody (Abcam, ab49129, 1:2000 dilution), anti-cardiac Troponin I (cTnI) (Thermo, MA1-20112, 1:250 dilutiion), anti-Troponin I1 (ssTnI) (abcam, EPR17120-11, 1:1000 dilution), anti-GAPDH (Cell Signaling, 5174, 1:1000 dilution) and anti-actin monoclonal antibody (Merck Millipore, MAB1501, 1:5000 dilution). The secondary antibodies used were ECL peroxidase-labeled anti-mouse antibody (GE Healthcare, NA931, 1:5000 dilution) and ECL peroxidase-labeled anti-rabbit antibody (GE Healthcare, NA934, 1:5000 dilution).

**TR-FRET assay.** The LanthaScreen TR-FRET PGC1a-binding assay (Thermo Fisher Scientific, PV4408) was performed according to the manufacturer's instructions. Serial concentrations of T112 and GSK4716 (Sigma) were incubated with 8 nM His-tagged ERRγ LBD and 10 nM fluorescein-PGC1a coactivator peptide for 2 h. The FRET signal was measured by excitation at 320 nm and emission

at 535 and 460 nm for fluorescein and terbium, respectively, using EnVision Multimode Plate Reader (PerkinElmer) and Envision Manager (PerkinElmer).

**Library preparation and RNA sequencing**. Total RNA was extracted using the miRNeasy Micro Kit (QIAGEN), and the cDNA synthesis and library construction were performed using TruSeq Standed Total RNA (Illumina) according to the manufacturer's instructions. Libraries were sequenced on a NextSeq 500 using High Output Kit ver. 2.0 and $1 \times 75$ bp mode. Reads were mapped to the human genome hg19 using STAR.

**Bioinformatics**. Principal component analysis (PCA) and density plot construction were performed using publicly available mouse single-cell RNA-seq data[4] in Python. The data were downloaded from the GNomEx database (272R, 274R, 275–292R, 439R, and 440R) and aligned to the reference genome GRCm38 using STAR. To calculate FPKM values, we used Cufflinks and GENCODE vM16. mESC-CP, mESC-CMs, hESC-D20, hESC-Y1, and heart tissue data were downloaded from the NCBI GEO databank (GSE47948, GSE62913, and GSE46224)[4,18,36]. For comparisons with mouse developmental data, we used the HomoloGene database and converted counts to FPKM. The raw data were loaded into RStudio (R visual script) for processing. Gene expression data for reading, exploring, and pre-processing were conducted using Bioconductor package *NOISeq* in order to perform data exploratory analysis and differential expression for RNA-Seq data. A hierarchical-derived cluster dendrogram was generated using *hclust* in *stats* package and *agnes* in *cluster* package. Distances were calculated using Manhattan city-block distance. *K*-means were computed with *kmeans* function. Distance and correlation matrices were computed and visualized using *get_dist* and *fviz_dist* included in *factoextra* package. *fviz_cluster* function was used to compute cluster scatter plots. Heatmaps were employed to cluster the expression data for DE genes using R script. Statistical analysis and visualization of functional profiles for genes and gene clusters for GO term analyses were performed using *DOSE* and *clusterProfiler* packages. Ingenuity pathway analysis (IPA) was used for upstream pathways analysis.

**Reverse transcription and quantitative PCR (qPCR)**. Total RNA was extracted using the miRNeasy Micro-Kit (QIAGEN), and purified RNA was reverse transcribed using the SuperScript™ VILO™ cDNA Synthesis Kit (Thermo Fisher Scientific). qPCR was performed with Taqman probes, and the samples were analyzed using a QuantStudio 7 Flex (Thermo Fisher Scientific). The Taqman probes we used are GAPDH (Hs99999905_m1) and TNNI3 (Hs00165957_m1). The expression of TNNI3 was normalized to the expression of GAPDH, and the relative expression level was calculated using the DMSO sample as a control.

**Mitochondria staining and functional assays**. For mitochondrial staining, day-23 EBs (#8-5-3) treated with compounds were dissociated with Liberase (Sigma-Aldrich) for 1 h at 37 °C, 5% $CO_2$ followed by 0.5 × TrypLE Select for 10 min at 37 °C, 5% $CO_2$. The dissociated cells were stained by the Mitochondrial Staining Kit (Abcam, ab176747) for 30 min at 37 °C, 5% $CO_2$ and analyzed using FACSAria Fusion (BD Biosciences). Regarding 409B2 and 1390C1, day-23 EBs were dissociated the same way as above, transfected with miRNA-208a-tagBFP switch and control EGFP and seeded on a fibronectin-coated six-well plate as previously described[19]. The next day, these cells were treated with 0.25% trypsin/EDTA for 3–4 min, stained using the Mitochondrial Staining Kit for 30 min and analyzed using FACSAria Fusion.

For the mitochondrial functional assays, 40,000 day-23 hiPSC-EmGFP+ CMs were seeded on fibronectin-coated seahorse assay wells. Six days after the seeding, the OCR was measured using a Seahorse Bioscience XF24 extracellular flux analyzer. We calculated the mitochondrial functions (basal and maximal respiration and ATP production) based on changes in OCR after addition of oligomycin, FCCP, or antimycin/Rotenone.

**EdU assay**. EdU assays were done using the Click-iT™ EdU Alexa Fluor™ 647 Flow Cytometry Assay Kit (Thermo Fisher Scientific) and performed according to the manufacturer's instructions.

**Motion vector analysis**. Dissociated hiPSC-CMs treated with compounds were seeded on a fibronectin-coated spot in a 35-mm glass bottom dish at $5$–$6 \times 10^4$ cells/5 µL. After 2 h, Stempro-34 medium supplemented with 1% L-glutamine, 50 µg/mL AA, 150 µg/mL Transferrin, 0.4 mM MTG, and 5 ng/mL VEGF was added and changed every 3 days, and the cells were incubated for 20–21 days. Spontaneous beating of monolayer CMs was recorded for 10 s at a frame rate of 150 fps and 2048 × 2048 pixels using the SI8000 Live Cell Motion Imaging System (Sony). For each sample, a region of interest was selected in the whole monolayer of beating cells. SI8000C Analyzer Software was used to analyze contraction, relaxation and conduction velocities.

**Patch clamp electrophysiology**. Day-16 EBs were dissociated with Liberase (Sigma-Aldrich) for 1 h at 37 °C, 5% $CO_2$ followed by Accumax (Innovative Cell Technologies) for 10 min at 37 °C, 5% $CO_2$. The dissociated cells were seeded on

fibronectin-coated glass coverslips (3 mm × 7 mm) at $1 \times 10^5$ cells/well in one well of a 24-well plate and cultured for 24 h in Stempro-34 medium supplemented with 1% L-glutamine, 50 µg/mL AA, 150 µg/mL Transferrin, 0.4 mM MTG, 5 ng/mL VEGF, and 10 µM Y-27632. The next day, the media was changed to the same media without Y-27632 but with DMSO or 3 µM T112. The cells were used for the recordings 6–13 days after the plating. Patch clamp measurements were performed on small groups of cells (3–10 cells) using the Axopatch 200B amplifier (Molecular Devices) at 5 kHz in the current clamp mode. APs were recorded from spontaneously contracting hiPSC-CMs using the gap-free protocol. All signals were digitized at 10 kHz using Digidata 1322 A and analyzed with pCLAMP 9.2 or 10.7 software (Molecular Devices). The cells were kept at 35–37 °C and perfused with modified Tyrode's solution containing NaCl (120 mM), KCl (5 mM), $CaCl_2$ (2 mM), $MgCl_2 \cdot 6H_2O$ (1 mM), $Na_2HPO_4$ (0.84 mM), $MgSO_4$ (0.28 mM), $KH_2PO_4$ (0.22 mM), $NaHCO_3$ (27 mM), and glucose (5.6 mM), pH 7.4. Patch pipettes were prepared from glass capillaries with tip resistances of 3–5 MΩ when filled with pipette solution using a micropipette puller (P-97/IVF, Sutter Instruments). The pipette solution consisted of 130 mM KOH, 130 mM L-aspartic acid, 20 mM KCl, 5 mM NaCl, 10 mM HEPES, 5 mM Mg-ATP, 10 mM EGTA, and 1 mM $MgCl_2$ adjusted to pH 7.2 with KOH. MDP, AP amplitude (APA), and maximum upstroke velocity (Vmax) were calculated from the average AP of 10 consecutive and stable waves. The minimum derivative value of each AP was calculated using OriginPro 2021 software. Notches were identified for derivative values <−3 as a cut-off set[29].

**Measurement of cell area**. Dissociated hiPSC-CMs were replated onto a CellCarrier-96 plate (PerkinElmer) at $1 \times 10^4$ cells/well. The cells were treated with DMSO or 3 µM T112 for 6 days and then fixed with 4% PFA. The cells were stained with cTNT (Thermo Fisher Scientific, MA5-12960, 1:500 dilution) and Hoechst. Imaging and analysis were performed using the Opera Phenix High Content Imaging System (PerkinElmer). Cells (<500 µm²) were removed as debris during the analysis.

**Measurement of cellular roundness**. Dissociated hiPSC-CMs treated with compounds were replated onto 20× FN-coated CellCarrier96 Ultra Microplates (Perkin Elmer) at $1 \times 10^4$ cells/well and incubated for 4 days. Then the cells were fixed with 4% PFA and stained with cTNT (Thermo, MS-295-P, 1:500 dilution) and goat anti-rabbit IgG (H + L) highly cross-adsorbed secondary antibody, Alexa Fluor 546 (Invitrogen, A-11035, 1:500 dilution). Images were acquired using an OperaPhenix imager (Perkin Elmer, Waltham, MA). Roundness was measured using Harmony High-Content Imaging and Analysis Software (Perkin Elmer).

**Transmission electron microscopy**. hiPSC-CMs were fixed at room temperature with 2.0% glutaraldehyde (Electron Microscopy Sciences) and 4.0% paraformaldehyde (Electron Microscopy Sciences) in 0.1 M sodium cacodylate buffer (CB) (Electron Microscopy Sciences) and then transferred to ice, where fixation was continued for 48 h. Post-fixation staining was performed with 1% osmium tetroxide (Electron Microscopy Sciences) in 0.15 M chilled CB for 30 min. The cells were rinsed five times 2 min each in 0.15 M chilled CB. The samples were then dehydrated in a cold graded ethanol (Sigma-Aldrich) series (50%, 70%, 90%, 100%, and 100%; 3 min each grade), rinsed once with anhydrous ethanol for 3 min at room temperature, and infiltrated in Durcupan ACM resin (Sigma-Aldrich) using 1:1 (v/v) anhydrous ethanol and resin for 30 min, 100% resin three times 1 h each, then in fresh resin, and polymerized in a vacuum oven at 60 °C for 48 h. Transmission electron microscopy was performed using a Hitachi H-7650 operated at 120 kV. The sarcomere length was calculated by measuring the distance between the Z lines.

**dbcAMP treatment**. ERRγ KO hiPSC-CMs were treated with 0.5 mM dbcAMP (Sigma-Aldrich) for 9 days (from days 8 to 17) or 7 days (from days 10 to 17).

**Statistics and reproducibility**. The data are presented as means and SEM. Statistical analysis was performed using GraphPad Prism (GraphPad software). Differences between experimental groups were analyzed by unpaired two-tailed *t*-test or one-way ANOVA followed by Dunnett's test or Tukey's honest significant difference (HSD) test. The micrographs shown in Fig. 1d, 3a and Supplementary Fig. 1e are representative images of one of the more than 100 successful differentiations. The images in Fig. 2b and 6a are representative images of more than 20 successful experiments. The TEM images in Fig. 5k and Supplementary Fig. 5a–f are representative images of more than 20 images under each condition from 2 consecutive experiments for each line. The fluorescence images in Supplementary Fig. 2b, c are images from a single experiment, but individual reagent conditions were run at least twice to confirm reproducibility. The fluorescence images in Supplementary Fig. 4a and g are representative images of three independent experiments. The micrographs and flow cytometry images shown in Supplementary Fig. 6d are representative images of three independent experiments.

**Reporting summary**. Further information on research design is available in the Nature Research Reporting Summary linked to this article.

## Data availability
The experimental data supporting the findings of this study are available from the corresponding author upon reasonable request. The RNA-seq data reported in this paper have been deposited in NCBI's Gene Expression Omnibus and are accessible through GEO series accession number GSE135319. The mouse single cell RNA-seq data are publicly available from the GNomEx database (accession numbers: 272R, 274R, 275–277R, 279–292R, 439R, and 440R). mESC-CP, mESC-CMs, hESC-D20, hESC-Y1, and heart tissue data are publicly available from the NCBI GEO databank (accession numbers: GSE47948, GSE62913, and GSE46224). Regarding western blots, source data are provided with this paper.

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

## Acknowledgements
We thank Drs. Shinya Yamanaka and Seigo Izumo for supporting this project. We also thank Takako Sono, Ayaka Sakoda, Miyuki Nomura, Yuko Ishikawa, and Haruka Yamamoto (T-CiRA) for preparing some of the materials and technical assistance, and Azusa Hama and Yoko Uematsu (CiRA), Shizuka Murakami, Yoshimitsu Nakashima, Reiko Oikawa (Takeda Pharmaceutical Company Limited), Takanori Matsuo and Aya Higashide (T-CiRA) for their administrative support. We also thank Peter Karagiannis (CiRA) for reading the manuscript. We used development samples provided free from Oji Holdings Corporation for the analysis. This research was supported by the Takeda–CiRA collaboration program, a grant from Takeda Pharmaceutical Company Limited, a grant from the Leducq Foundation (18CVD05), a grant from SECOM Science and Technology Foundation, grants from the Research Center Network for Realization of Regenerative Medicine (JP19bm0104001, JP19bm0204003, JP19bm0804008, and JP20bm0804022), Research on Regulatory Science of Pharmaceuticals and Medical Devices (JP19mk0104117), and Research Project for Practical Applications of Regenerative Medicine (JP19bk0104095) provided by the Japan Agency for Medical Research and Development, JSPS KAKENHI Grants (JP18K15120, JP18KK0461, JP19K16041 and JP17H04176) and the iPS Cell Research Fund. Kyoto University and Takeda Pharmaceutical Company Limited have filed a patent application broadly relevant to this work.

## Author contributions
K.M., K.D., M.N.-K., N.T., K.I., T.N., and Y.Y. conceived and designed the project. K.M., K.D., M.N.-K., S.K., Y.F., M.S., Y.N., I.T. and S.C.N. performed the experimental work. T.S. and S.I. performed the CRISPR constructions. M.S., A.L.-C., C.O., M.N., T.M., and T.A. analyzed the RNA-seq data. T.H. and Y.F. performed the transmission electron microscopy work. K.M., K.D., M.N.-K., A.L.-C., T.N., and Y.Y. wrote the manuscript. All authors discussed the results.

## Competing interests
Y.Y. received research expense from Takeda Pharmaceutical Company, Ltd, and K.D., S.K., S.C.N., S.I, T.M., T.A., N.T., K.I., T.N. are employees of Takeda Pharmaceutical Company, Ltd. K.M., S.K., and Y.Y. are the inventors of the patent application (WO2019/189554). The other authors declare no competing interests.
