## [Peer Review File · Nature Communications]

Reviewers' comments:

Reviewer #1 (Remarks to the Author):

This is an interesting paper seeking to employ the discovery of small molecules in order to advance the maturation of hiPSC-derived cardiac muscle in dish.

To enable this work, they developed an interesting platform to "screen" for maturation agents via gene editing hiPSCs. They engineered into the TNNI1 locus GFP and in the same line in frame into the TNNI3 locus mCherry fluorescent reporters.

With this system established, they screened a small molecule library and identified small molecules that modulate two pathways - activation of estrogen-related receptor gamma (ERR γ) in combination with an inhibitor of S-phase kinase-associated 36 protein 2 (SKP2) that provide enhancement in TNNI3 maturation marker in hiPSC-CMs.

This system is of interest to the field.

Main comments.

1. The screening tool they have developed is interesting and appears well positioned as an effective means to discover small molecules to advance hiPSC-CM maturation. The TNNI genetic switch is well suited to track maturation, as they have indicated. That said, their use of fluorescent reports offers only a qualitative analysis of relative gene induction. To complement the screen, it will be important to follow up with quantitative analysis via Western blot for TnI to directly determine the cTnI/ssTnI ratio in hiPSC-CMs (as described in ref. 8) in response to the different small molecule maturation combinations used. This is important because the present data using fluorescence (e.g., fig. 3a /combo) shows both TNNI GFP and TNNI3 mCherry fluorescence increasing in the combo group. This could be attributed to simply an increase in sarcomere content formation (more myofibril content) with the same cTnI/ssTnI ratio. Quantitative Western blots directly comparing in each well the cTnI/ssTnI ratio (as shown in ref. 8) would definitively establish if stoichiometric maturation is occurring with adult cTnI actually replacing the immature ssTnI protein.

2. Greater details of the small molecule library are required. Ln 128: "We extensively screened a library of 9,048 chemical compounds ..." What library was used? Why was this rather small library selected? - what criteria were used in selecting this one? More details are needed here. Also, why not use a larger more extensive library covering a larger chemical space?

3. In a similar vein, Ln 123: "...we tested 12 commercially available compounds reported to facilitate cardiac maturation..." What were these 12 compounds? What data supports that these advance maturation? - references should be provided to support these 12.

4. Fig. 4b. Is the green - red key backwards? --- with -3 to 3 switched --- green being 3 (high) and red -3 (low)?

Minor.

1. The term "monotherapy" seems out of place here and should be replaced with small molecule or similar.

Reviewer #2 (Remarks to the Author):

Yoshida and colleagues describe a new tool to monitor the TNNI isoform switch that occurs during the maturation of fetal to postnatal cardiomyocytes. hPSC-derived cardiomyocytes are generally functionally, genetically and structurally immature and a method that promotes all of these aspects of maturation would be welcome in the field. The reporter hiPSC line as such is a valuable screen to look for maturation factors which could be molecular or biophysical as recently shown in papers from Mills et al and Ronaldson-Bouchard et al. Here the claim is that just two molecular pathways are sufficient to induce maturation. Whilst this is potentially an interesting finding, there are multiple aspects of the paper that fall short of supporting the conclusions, not least a full analysis of the relevant maturation parameters. In addition, there is a considerable body of literature on the role of the ERRgamma pathway on maturation (cited as 22-25 by the authors), including overexpression data, which somewhat limits enthusiasm on novelty,

A number of specific points that need addressing are described below:

1. Why did the authors not combine the new factors with the two others that served as their positive control?
2. Analysis of a single clone expressing the reporter is very small number on which to base fairly strong conclusions. If a second reporter clone is not available, then at a minimum, the factors should be tested in two (other) independent hiPSC lines (not just one) and the extent of functional, structural and metabolic maturity determined as in the reporter line.
3. Line 115: please show the data (qPCR and immunofluorescence of corresponding protein) corresponding to this statement "The expression kinetics of EmGFP and mCherry mirrored the expression of TNNI1 and TNNI3, respectively.
4. Aspects of structural maturation appear to be induced by ERRgamma combined with cell cycle inhibition but there is no mention of T-tubules (ultrastructure, BIB1 expression), calveoli etc. Ultrastructural analysis is necessary to claim maturation. The authors mention in the introduction that "hallmarks represent common denominators of adult myocardium in humans: (1) efficient conversion of energy, (2) excitation-contraction coupling, (3) positive force-frequency relationship (4) anisotropy driven by cellular alignment and (5) achievement of adult cardiomyocyte-like electrophysiological property with increased conduction velocity. Of all of these functional parameters, they have actually based their conclusions essentially on (1). The authors need to include these parameters as well to substantiate their claim.

In sum, the paper has the basis to become interesting since it does claim maturation in the absence of physical constraint or pacing but the parameters selected for analysis are insufficient to support the claim. This needs to be addressed as described above.

Point By Point Response:

Reviewer #1:

This is an interesting paper seeking to employ the discovery of small molecules in order to advance the maturation of hiPSC-derived cardiac muscle in dish.

To enable this work, they developed an interesting platform to “screen” for maturation agents via gene editing hiPSCs. They engineered into the TNNI1 locus GFP and in the same line in frame into the TNNI3 locus mCherry fluorescent reporters.

With this system established, they screened a small molecule library and identified small molecules that modulate two pathways - activation of estrogen-related receptor gamma (ERR γ) in combination with an inhibitor of S-phase kinase-associated 36 protein 2 (SKP2) that provide enhancement in TNNI3 maturation marker in hiPSC-CMs.

This system is of interest to the field.

Response:

We thank the reviewer for the positive feedback and addressed the specific comments below.

Main comments.

1. *The screening tool they have developed is interesting and appears well positioned as an effective means to discover small molecules to advance hiPSC-CM maturation. The TNNI genetic switch is well suited to track maturation, as they have indicated. That said, their use of fluorescent reports offers only a qualitative analysis of relative gene induction. To complement the screen, it will be important to follow up with quantitative analysis via Western blot for TnI to directly determine the cTnI/ssTnI ratio in hiPSC-CMs (as described in ref. 8) in response to the different small molecule maturation combinations used. This is important because the present data using fluorescence (e.g., fig. 3a /combo) shows both TNNI GFP and TNNI3 mCherry fluorescence increasing in the combo group. This could be attributed to simply an increase in sarcomere content formation (more myofibril content) with the same cTnI/ssTnI ratio. Quantitative Western blots directly comparing in each well the cTnI/ssTnI ratio (as shown in ref. 8) would definitively establish if stoichiometric maturation is occurring with adult cTnI actually replacing the immature ssTnI protein.*

Response:

We thank the reviewer for the suggestion. According to the reviewer’s comment, we have included quantitative data of the cTnI/ssTnI ratio in Figure 5c and Supplementary Figure 4d. These data clearly show that T112 (ERR γ agonist) significantly increased the cTnI/ssTnI ratio, indicating the maturation-induced isoform switch of TNNI.

2. *Greater details of the small molecule library are required. Ln 128: “We extensively screened a library of 9,048 chemical compounds ...” What library was used? Why was this rather small library selected? – what criteria were used in selecting this one? More details are needed here. Also, why not use a larger more extensive library covering a larger chemical space?*

Response:

We added more detailed explanation about the library in the section “High throughput screening” in the Methods. We agree that a larger library is preferred to search for more effective compounds. However, the library we used was designed to cover a diverse and broad chemical space and was biologically annotated, as we explain in the revised manuscript (Supplemental Figure 2d). It is possible that modifying the molecular formula of the hit compounds in this study will lead to more highly active compounds, which is our future plan for synthesizing novel compounds. Accordingly, we added the following sentences to the main text (line 555-567)

“The library was designed to cover a diverse chemical space and included elements from a biologically annotated compound library, focused kinase library and internal/external repositioning library at Takeda Pharmaceutical Company Ltd. The criteria for the compound selection were in vitro pharmacological activity on individual target proteins with an EC50 or IC50 value less than or equal to 1 μM. The activity was determined by cell-free and cell-based binding and functional assays, as shown in Supplementary Figure 2d. The resultant compound library interacted with 1,804 unique proteins. Furthermore, 70% of these target proteins were annotated by multiple compounds, which must be taken into account in order to avoid misinterpretation of the experimental results due to possible off-target effects by the small molecules. After the initial refinement, close analogues of an in-house extensive compound library were tested by an hiPSC-based assay to validate any identifiable structure-activity relationship (SAR).”

3. In a similar vein, Ln 123: “...we tested 12 commercially available compounds reported to facilitate cardiac maturation...” What were these 12 compounds? What data supports that these advance maturation? – references should be provided to support these 12.

Response:

We thank the reviewer for the comment. The 12 compounds are described in reference #8 (Bedada et al *Stem Cell Reports* 3, 594-605 (2014)). We have added a description to the revised manuscript (line 118-line 120) as below.

“As a positive control for TNNI3 expression, we tested 12 commercially available compounds reported to facilitate cardiac maturation both as single compounds and in combinations⁸.”

#8 reference: Bedada FB, et al. Acquisition of a quantitative, stoichiometrically conserved ratiometric marker of maturation status in stem cell-derived cardiac myocytes. *Stem Cell Reports* 3, 594-605 (2014).

4. Fig. 4b. Is the green – red key backwards? --- with -3 to 3 switched --- green being 3 (high) and red -3 (low)?

Response:

Thank you for pointing this error out. It has been corrected in the revised manuscript (green -3 (low) to red 3 (high)).

Minor comment

1. *The term “monotherapy” seems out of place here and should be replaced with small molecule or similar.*

Response:

We thank the reviewer for the comment. We have replaced “monotherapy” with “single compound”.

Reviewer #2:

Yoshida and colleagues describe a new tool to monitor the TNNI isoform switch that occurs during the maturation of fetal to postnatal cardiomyocytes. hPSC-derived cardiomyocytes are generally functionally, genetically and structurally immature and a method that promotes all of these aspects of maturation would be welcome in the field. The reporter hiPSC line as such is a valuable screen to look for maturation factors which could be molecular or biophysical as recently shown in papers from Mills et al and Ronaldson-Bouchard et al. Here the claim is that just two molecular pathways are sufficient to induce maturation. Whilst this is potentially an interesting finding, there are multiple aspects of the paper that fall short of supporting the conclusions, not least a full analysis of the relevant maturation parameters. In addition, there is a considerable body of literature on the role of the ERRgamma pathway on maturation (cited as 22-25 by the authors), including overexpression data, which somewhat limits enthusiasm on novelty. A number of specific points that need addressing are described below:

Response:

We wish to express our appreciation to the Reviewer for his or her insightful comments, which have helped us significantly improve the paper.

Comments

1. *Why did the authors not combine the new factors with the two others that served as their positive control?*

Response:

We thank the reviewer for the comment. We have included the data of TNNI3-mCherry intensity in hiPSC-CMs treated with our compounds with or without positive controls. The co-treatment with Combo and PC (Combo+PC) significantly increased the signal intensity compared to PC alone. However, we found no significant difference between Combo and Combo+PC (Supplemental Figure 3b). Accordingly, we added the following sentences to the main text (line 169-170)

“However, no significant difference was observed in mCherry intensity between Combo with or without the positive control (Supplementary Figure 3b).”

2. *Analysis of a single clone expressing the reporter is very small number on which to base fairly strong conclusions. If a second reporter clone is not available, then at a minimum, the factors should be tested in two (other) independent hiPSC lines (not just one) and the extent of functional, structural and metabolic maturity determined as in the reporter line.*

Response:

We thank the reviewer for the suggestion. We assessed the CM maturation by the compounds using two hiPSC lines, 409B2 and 1390C1, as well as our original double reporter hiPSC line (#8-5-3). We include the mitochondrial data (metabolic maturation) and TNNI3 expression (sarcomeric maturation) of the two hiPSC lines in the revised manuscript (Supplementary Figure 4a, b and c). Further, we also include data of 1390C1 regarding the western blot analysis of TNNI3 and the cTnI/ssTnI ratio (Supplementary Figure 4c and d), cell area (Supplementary Figure 4e), sarcomere length (Supplementary Figure 4f), contractile properties (Supplementary Figure 4i) and ultrastructure (Supplementary Figure 5d-f) as well as the original hiPSC line (#8-5-3). These data demonstrated that the ERR γ agonist enhances sarcomere and metabolic maturation in multiple hiPSC lines.

3. *Line 115: please show the data (qPCR and immunofluorescence of corresponding protein) corresponding to this statement “The expression kinetics of EmGFP and mCherry mirrored the expression of TNNI1 and TNNI3, respectively.*

Response:

We have included the data showing a significant correlation between the intensity of EmGFP and mCherry and the expression levels of TNNI1 and TNNI3, respectively, in the revised manuscript (Supplemental Figure 1f). Accordingly, we added the following sentences to the main text (line 110-112)

“The expression kinetics of EmGFP and mCherry mirrored the expression of TNNI1 and TNNI3, respectively (Supplementary Figure 1f).”

4. *Aspects of structural maturation appear to be induced by ERR γ combined with cell cycle inhibition but there is no mention of T-tubules (ultrastructure, BIB1 expression), calveoli etc. Ultrastructural analysis is necessary to claim maturation. The authors mention in the introduction that “hallmarks represent common denominators of adult myocardium in humans: (1) efficient conversion of energy, (2) excitation-contraction coupling, (3) positive force-frequency relationship (4) anisotropy driven by cellular alignment and (5) achievement of adult cardiomyocyte-like electrophysiological property with increased conduction velocity. Of all of these functional parameters, they have actually based their conclusions essentially on (1). The authors need to include these parameters as well to substantiate their claim. In sum, the paper has the basis to become interesting since it does claim maturation in the absence of physical constraint or pacing but the parameters selected for analysis are insufficient to support the claim. This needs to be addressed as described above.*

Response:

We thank the reviewer for the suggestion. To further characterize the maturation induced by the compounds, we have included the following data.

- cTnI/ssTnI ratio: Figure 5c and Supplementary Figure 4d
- Cellular roundness (anisotropy): Figure 5e
- Contractile properties and conduction velocity measured by motion vector analysis: Figure 5g and Supplementary Figure 4i
- Electrical properties (patch clamp): Figure 5h
- Ultrastructure: Figure 5i and Supplementary Figure 5a-f

The transmission electron microscopy data demonstrated that hiPSC-CMs treated with the ERR γ agonist displayed T-tubules, which are critical for the synchronous calcium-induced calcium release that underlies efficient excitation–contraction coupling, as well as increased sarcomere length. Furthermore, we confirmed the compounds induce cellular anisotropy in hiPSC-CMs, and motion vector analysis showed that the ERR γ agonist significantly increased contraction velocity, relaxation velocity and propagation velocity, which suggested that the agonist-treated CMs have enhanced contractility and electrical connection by gap junctions. In addition, patch clamp data showed the agonist significantly increased APA and V_{max} in hiPSC-CMs, and RNA-seq data showed that the expression profile of electrophysiological properties in agonist-treated CMs is similar to that in adult ventricular CMs. Taken together, our data indicate that the agonist significantly enhanced cardiac maturation in aspects of sarcomere, metabolism, contractility and electrophysiology.

In this study, we demonstrated CM maturation can be achieved by adding just a single ERR γ agonist to the CMs without any electrical or mechanical stimulation. Of note, T-tubule formation in CMs in conventional monolayer culture, which was not reported previously to our knowledge, is induced by our compound. Currently, we are constructing a 3D microtissue model to further evaluate the CM maturation, including the force–frequency relationship, for a future paper.

Once again, we thank the reviewers for their feedback. Due to the outbreak of COVID-19, our research activity was substantially restricted. Nonetheless, we hope all suggestions made by the reviewers have been addressed by the additional data and revisions to the manuscript. Thank you again for your consideration of our manuscript for publication in *Nature Communications*.

Sincerely yours,

REVIEWER COMMENTS

Reviewer #1 (Remarks to the Author):

The revised paper by Miki et al., is improved; however, they have left some of the initial key questions still unanswered. The premise of the paper remains strong and the hiPSC lines developed should be of interest to the field.

Main concern.

1. Overall they need to temper the wording on the effectiveness of the small molecules they have identified as at best, from the stated position of using TnI isoform to track cardiac muscle maturation, the data support a transition from embryonic/fetal-like to neonatal-like TnI isoform profile and not to an adult-like state. This needs to be clearly stated in the abstract and concluding statements in the paper to accurately reflect the findings.

Specifically:

1a. Western blots were performed to calculate the cTnI/ssTnI isoform ratio, which is central to the paper. However, rather than using a pan TnI isoform antibody with equal avidity for both cTnI and ssTnI they used two different TnI isoform antibodies. No details or references are provided on the relative specificity of these antibodies, and this needs to be included. Also, calculating a TnI isoform ratio (as done in ref. 8) becomes more difficult especially as they provide no controls including neo/adult heart samples showing the TnI ratio transition during heart development.

1b. Of similar concern are the W. blots shown Fig. 5C vs Supple 4d wherein ssTnI appears with single or double bands? Which is ssTnI? Molecular weight markers and positive controls for ssTnI and cTnI should be provided in these gels to allow for direct analysis.

1c. Presuming 1a and 1b can be addressed, there is a large concern that the statistically significant increase in cTnI isoform content is not "physiologically" significant in terms of maturation toward the adult cTnI/ssTnI ratio, which is 100% (ref. 8). Rather, these data show specific drug cocktails can increase cTnI isoform protein content to about 10-15% cTnI/ssTnI, which makes these neonatal-like rather than adult-like. Page 12, the phrase "cTnI isoform proteins were highly expressed in hiPSC-CMs..." is not accurate. Should be "...modestly increased to resemble a neonatal myocardium-like cTnI/ssTnI ratio."

2. Authors' responses to the request for the details/specifics of the 12 commercially available compounds tested was not sufficient. The paper cited by Bedada (ref 8) does not study or list 12 compounds (only T3 in my reading of that paper). They need to list the 12 compounds and concentrations they used in this paper to enable others the methods in context to examine these in the future. These can easily be detailed in the supplement section of the paper.

Reviewer #2 (Remarks to the Author):

The authors have addressed several of my questions acceptably, most notable the request to include cardiomyocytes from more independent hiPSC lines and to extend the parameters to a full set of those normally associated with cardiomyocyte maturation. There is still some quantification of these new data missing: % CMs with T-tubules for example, als % with the typical "mature" notch in the action potential. Although the screening method is nice, the question of how novel the outcome is, is not really addressed by the authors despite the comment in the review. Even without a screen, there is substantial data in the (mouse and human) literature that implicates ERR activators as candidates, much like Dexamethasone binding to another nuclear receptor was identified empirically previously.

Also during the period of revision, several studies have been published in this area on hiPSC-CM maturation, notably doi: [10.1016/j.stem.2020.05.004](https://doi.org/10.1016/j.stem.2020.05.004) which showed that dbcAMP could induce electrophysiological maturation and that all the features on maturation described here were also seen when various cellular co-cultures were used to increase intracellular cAMP in hiPSC-CMs. No pacing or scaffold proteins were required. In fact there is some data showing that cAMP actually regulates ERR expression. This needs discussing in the paper and relevant additional references included, other than Ronaldson-Bouchard et al and Tilburcy et al. Of note, an extensive correction has been published on the Ronaldson-Bouchard paper and there are comments of concern on PubPeer about the data.

Response to the comments by Reviewer #1:

The revised paper by Miki et al., is improved; however, they have left some of the initial key questions still unanswered. The premise of the paper remains strong and the hiPSC lines developed should be of interest to the field.

Main concern.

1. Overall they need to temper the wording on the effectiveness of the small molecules they have identified as at best, from the stated position of using TnI isoform to track cardiac muscle maturation, the data support a transition from embryonic/fetal-like to neonatal-like TnI isoform profile and not to an adult-like state. This needs to be clearly stated in the abstract and concluding statements in the paper to accurately reflect the findings.

Response:

We thank the reviewer for the suggestion and agree with the point. To accurately reflect our findings, we have changed the wording from adult to neonatal. Accordingly, the following modifications were made.

- line 31-37: “Notably, expression, morphological, functional, and molecular analyses indicated that hiPSC-CMs treated with the ERR γ agonist showed a larger cell size, longer sarcomere length, the presence of transverse tubules, and enhanced metabolic function and contractile and electrical properties, which together suggest that ERR γ -treated hiPSC-CMs have a mature cellular property consistent with neonatal CMs.”
- line 270-276: “However, hiPSC-CMs treated with T112 or Combo expressed ssTnI (TNNI1), which is unlike adult heart. The cTnI/ssTnI protein isoform switch in mouse ventricular heart begins irreversibly during the neonatal period, and the switch eventually becomes 100% cTnI in the adult hearts of all mammalian species including humans⁸. Thus, our data suggested that hiPSC-CMs treated with T112 or Combo had the maturity of neonatal CMs.
- line 464-468: “Nonetheless, our study depicts a major advancement in the field by providing a short and scalable method that generates in vitro hiPSC-CMs displaying mature features sufficiently to reliably model diseases and undergo drug screening with a high resemblance to functional matured CMs.
- Figure 7: We modified the Schematic representation.

Specifically:

1a. Western blots were performed to calculate the cTnI/ssTnI isoform ratio, which is central to the paper. However, rather than using a pan TnI isoform antibody with equal avidity for both cTnI and ssTnI they used two different TnI isoform antibodies. No details or references are provided on the relative specificity of these antibodies, and this needs to be included. Also, calculating a TnI isoform ratio (as done in ref. 8) becomes more difficult especially as they provide no controls including neo/adult heart samples showing the TnI ratio transition during heart development.

Response:

We thank the reviewer for the suggestion. Accordingly, we have included the marker and data of an adult heart sample in Figure 5c and supplementary Figure 4d. We were unable to include a sample of the neonatal heart due to unavailability. Regarding the specificity of both antibodies, as shown in Figure 5c and supplementary Figure 4d, we confirmed their specific detection of cTnI and ssTnI using adult heart sample and hiPSC-CMs (immature CMs), finding each antibody showed specific bands at the expected size. We also included information of the manufacturer and catalog number in the Methods section and Reporting Summary.

In this study, the reason why we did not use pan-TnI antibody is that we used hiPSCs with T2A-EmGFP sequence inserted into one allele of the ssTnI locus (#8-5-3 line). In cardiomyocytes derived from the reporter iPSC line, T2A peptides (17 peptides) are added to the C-terminus of ssTnI, therefore, two bands of ssTnI were detected in the western blot (at about 22 kDa and 24 kDa), as shown in figure A and B below. We noticed that the large sized band overlaps with the band of cTnI (24 kDa). Since pan-TnI antibody could not distinguish cTnI and ssTnI-T2A, we did not use it. In 1390C1, which does not have the T2A-reporter sequence, we could detect the cTnI band in only the T112 and Combo groups using pan-TnI antibody, as shown in figure C below.

Figure A

ssTnI specific Ab

#8-5-3
TNNI1-T2A-EmGFP

Figure B

pan-TnI Ab

#8-5-3
TNNI1-T2A-EmGFP

Figure C

pan-TnI Ab

1390C1

1b. Of similar concern are the W. blots shown Fig. 5C vs Supple 4d wherein ssTnI appears with single or double bands? Which is ssTnI? Molecular weight markers and positive controls for

ssTnI and cTnI should be provided in these gels to allow for direct analysis.

Response:

We thank the reviewer for the comment. As the reviewer pointed out, there are double bands for ssTnI in Figure 5c. In this study, we used hiPSCs containing an EGFP cassette connected to the TNNI1 (ssTnI) locus by a T2A sequence. As mentioned above, T2A peptides (17 peptides) are added to the C-terminus of ssTnI. Thus, we are certain that the longer band is ssTnI with the T2A product. In fact, we did not observe the longer band in the western blot of another hiPSC line without reporter integration (supplementary Figure 4d).

We have accordingly included the following sentence to the Figure 5c legend.

line 879-880

“For ssTnI, two bands were detected: the shorter one is wild type ssTnI and the longer one is ssTnI with T2A peptides.”

1c. Presuming 1a and 1b can be addressed, there is a large concern that the statistically significant increase in cTnI isoform content is not “physiologically” significant in terms of maturation toward the adult cTnI/ssTnI ratio, which is 100% (ref. 8). Rather, these data show specific drug cocktails can increase cTnI isoform protein content to about 10-15% cTnI/ssTnI, which makes these neonatal-like rather than adult-like. Page 12, the phrase “cTnI isoform proteins were highly expressed in hiPSC-CMs...” is not accurate. Should be “...modestly increased to resemble a neonatal myocardium-like cTnI/ssTnI ratio.”

Response:

We thank the reviewer for the comment. We have changed the sentence to the following. (Deleted words are shown with strikethrough lines, and new sentence are underlined.)

line 265-276

“Western blot analysis showed that cTnI isoform proteins were ~~highly~~ expressed in hiPSC-CMs treated with T112 or Combo and that the cTnI/ssTnI protein isoform ratio was ~~significantly~~ increased in both groups (Figure 5c). In addition, increases of the TNNI3 expression and cTnI/ssTnI protein isoform ratio in hiPSC-CMs treated with T112 or Combo were observed in other hiPSC lines (409B2 and 1390C1) (Supplementary Figure 4c and d). However, hiPSC-CMs treated with T112 or Combo expressed ssTnI (TNNI1), which is unlike adult heart. The cTnI/ssTnI protein isoform switch in mouse ventricular heart begins irreversibly during the neonatal period, and the switch eventually becomes 100% cTnI in the adult hearts of all mammalian species including humans⁸. Thus, our data suggested that hiPSC-CMs treated with T112 or Combo had the maturity of neonatal CMs.”

2. Authors’ responses to the request for the details/specifics of the 12 commercially available compounds tested was not sufficient. The paper cited by Bedada (ref 8) does not study or list 12 compounds (only T3 in my reading of that paper). They need to list the 12 compounds and concentrations they used in this paper to enable others the methods in context to examine these in the future. These can easily be detailed in the supplement section of the paper.

Response:

We thank the reviewer for the comment and sincerely apologize. We realized we wrote the incorrect number. The correct number is 11, and we have corrected the manuscript accordingly. Regarding the 12 compounds and their concentrations, we have listed them in supplementary Figure 2a right and added the following sentences to main text (line 120-123).

“As a positive control for TNNI3 expression, we tested 12 commercially available compounds reported to facilitate cardiac maturation both as single compounds and in combinations¹¹ (Supplementary Figure 2a).”

#11 reference: Foldes G, et al. Aberrant alpha-adrenergic hypertrophic response in cardiomyocytes from human induced pluripotent cells. *Stem Cell Reports* 3, 905-914 (2014).

Response to the comments by Reviewer #2:

The authors have addressed several of my questions acceptably, most notable the request to include cardiomyocytes from more independent hiPSC lines and to extend the parameters to a full set of those normally associated with cardiomyocyte maturation. There is still some quantification of these new data missing: % CMs with T-tubules for example, als % with the typical "mature" notch in the action potential.

Response:

We thank the reviewer for the suggestion. We have included the data of notch in the action potential (Figure 5i and Supplementary Figure 4j). As shown in these figures, the number of action potentials with notch clearly increased in T112-treated CMs compared with DMSO-treated CMs. These patch clamp data including APA and Vmax showed that T112 enhances electrophysiological maturation in hiPSC-CMs.

We also considered quantifying the percentage of cells with T-tubes. However, since only a limited region of a cell is displayed in one electron microscope image (Figure 5k and Supplementary Figure 5), it is very difficult to determine whether each cell has T-tubules or not and observe enough cells to provide quantitative data. Thus, these data are not in the revised manuscript. Actually, no previous study shows quantitative data as far as we know.

Although the screening method is nice, the question of how novel the outcome is, is not really addressed by the authors despite the comment in the review. Even without a screen, there is substantial data in the (mouse and human) literature that implicates ERR activators as candidates, much like Dexamethasone binding to another nuclear receptor was identified empirically previously.

Response:

In our screening panel, we found a novel ERR γ agonist that is completely different from commercially available ERR γ agonists. We believe this discovery was not possible by simply examining commercially available drugs or by surveying the existing literature. As shown in Supplementary Figure 6d, the commercially available ERR γ agonist GSK4716 failed to increase

TNNI3 expression. These findings revealed the idea that higher agonistic activity than that of conventional ERR γ agonists is required to induce CM maturation with TNNI3 upregulation.

In addition, our compound identified in the screening is about 100 times more active than commercially available compounds, which not only provides new possibilities for cardiac maturation in hiPSC-CMs, but also has potential applications in other cell types and in vitro studies on ERR γ . We have included these points in the Discussion section (line 373-386).

“In the present study, we found a novel ERR γ agonist (T112) from our screening panel with a different chemical structure from the commercially available ERR γ agonist (GSK4716) and revealed that it increased mitochondria mass and function in hiPSC-CMs as well as TNNI3 upregulation (Figure 5a-b, Supplementary Figure 4a and b). In contrast, GSK4716 failed to increase TNNI3 expression (Supplementary Figure 6d). A LanthaScreen TR-FRET coactivator assay showed that the effective concentrations 50 (EC₅₀) of GSK4716 and T112 are 2.9×10^{-6} M and 3.0×10^{-8} M, respectively (data not shown), indicating GSK4716’s weaker activity. These results indicate that ERR γ agonistic activity can enhance the cardiac maturation of hiPSC-CMs in a dose-dependent manner with a threshold defined by the effective concentration inducing TNNI3 expression. In addition, T112 has potential applications to other cell types and for the in vitro study of ERR γ .”

Also during the period of revision, several studies have been published in this area on hiPSC-CM maturation, notably doi: 10.1016/j.stem.2020.05.004 which showed that dbcAMP could induce electrophysiological maturation and that all the features on maturation described here were also seen when various cellular co-cultures were used to increase intracellular cAMP in hiPSC-CMs. No pacing or scaffold proteins were required. In fact there is some data showing that cAMP actually regulates ERR expression. This needs discussing in the paper and relevant additional references included, other than Ronaldson-Bouchard et al and Tilburcy et al. Of note, an extensive correction has been published on the Ronaldson-Bouchard paper and there are comments of concern on PubPeer about the data.

Response:

As mentioned by the reviewer, we have included the reference (doi: 10.1016/j.stem.2020.05.004) and discussed the study in the Discussion section (line 441-451). In addition, we investigated the addition of dbcAMP to ERR γ KO iPSC-CMs (Supplementary Figure 6e). We found dbcAMP did not increase the expression of TNNI3-mCherry in ERR γ KO hiPSC-CMs, suggesting that ERR γ might contribute to the cAMP-mediated cardiac maturation process. The finding is in consistent with previous literature showing that cAMP regulates ERR γ expression.

Once again, we thank the reviewers for their feedback. We hope all suggestions made by the reviewers have been addressed by the additional data and revisions to the manuscript. Thank you again for your consideration of our manuscript for publication in *Nature Communications*.

REVIEWERS' COMMENTS

Reviewer #1 (Remarks to the Author):

The authors have addressed well my concerns.